# Tonotopy is not preserved in a descending stage of auditory cortex

Miaoqing Gu[1,2†], Shanshan Liang[3†], Jiahui Zhu[2], Ruijie Li[3], Ke Liu[3], Xuanyue Wang[3], Frank W Ohl[4,5,6], Yun Zhang[7], Xiang Liao[8], Chunqing Zhang[9], Hongbo Jia[1,4,10], Yi Zhou[11]*, Jianxiong Zhang[3]*, Xiaowei Chen[3,12]*

[1]School of Physical Science and Technology, Guangxi University, Nanning, China; [2]Guangxi Key Laboratory of Special Biomedicine and Advanced Institute for Brain and Intelligence, School of Medicine, Guangxi University, Nanning, China; [3]Brain Research Center and State Key Laboratory of Trauma and Chemical Poisoning, Third Military Medical University, Chongqing, China; [4]Leibniz Institute for Neurobiology (LIN), Magdeburg, Germany; [5]Insitute of Biology (IBIO), Otto-von-Guericke University, Magdeburg, Germany; [6]Center for Behavioral Brain Sciences (CBBS), Magdeburg, Germany; [7]State Key Laboratory of Structural Chemistry, Fujian Institute of Research on the Structure of Matter, Chinese Academy of Sciences, Fuzhou, China; [8]School of Medicine, Center for Neurointelligence, Chongqing University, Chongqing, China; [9]Institute of Brain and Intelligence, Third Military Medical University, Chongqing, China; [10]Brain Research Instrument Innovation Center, Suzhou Institute of Biomedical Engineering and Technology, Chinese Academy of Sciences, Suzhou, China; [11]Department of Military Cognitive Psychology, School of Psychology, Third Military Medical University, Chongqing, China; [12]LFC Laboratory (Chongqing Key Laboratory of Brain and Aerospace Intelligence) and Chongqing Institute for Brain and Intelligence, Guangyang Bay Laboratory, Chongqing, China

*For correspondence:
yzhou@tmmu.edu.cn (YZ);
jianxiong_zhang1988@tmmu.edu.cn (JZ);
xiaowei_chen@tmmu.edu.cn (XC)

†These authors contributed equally to this work

## eLife Assessment

This revised manuscript presents an **important** characterization of mouse auditory cortex receptive field organization, utilizing two-photon imaging of specific subpopulations. They demonstrate a degradation of tonotopic organization from the input to the output neurons. The strength of the evidence is **convincing**.

**Abstract** Previous studies based on layer specificity suggest that ascending signals from the thalamus to the sensory neocortex preserve spatially organized information, but it remains unknown whether sensory information descending from sensory neocortex to the thalamus also maintains such spatial organization pattern. By focusing on projection specificity, we mapped the tone response properties of two groups of cortical neurons in the primary auditory cortex (A1), based on the relationship between their specific connections to other regions and their function in ascending (thalamocortical recipient [TR] neurons) or descending (corticothalamic [CT] neurons) auditory information. A clear tonotopic gradient was observed among TR neurons, but not CT neurons. Additionally, CT neurons exhibited markedly higher heterogeneity in their frequency tuning and had broader bandwidth than TR neurons. These results reveal that the information flow descending from A1 to the thalamus via CT neurons is not arranged tonotopically, suggesting that the descending information flow possibly contributes to higher-order feedback processing of diverse auditory inputs.

## Introduction

In the mammalian auditory system, one of the most prominent features is the tonotopic organization – a spatially ordered gradient of neuronal frequency preference (*Bizley et al., 2005*; *Kajikawa et al., 2005*; *Merzenich et al., 1976*; *Morel et al., 1993*; *Nelken et al., 2004*; *Reale and Imig, 1980*; *Stiebler et al., 1997*; *Tani et al., 2018*). Clear tonotopic maps have been found in the auditory cortex (AuC) of many mammalian species, including humans (*Clopton et al., 1974*; *Humphries et al., 2010*), macaque monkeys (*Hackett and Stepniewska, 1998*), cats (*Lee et al., 2004*), ferrets (*Bizley et al., 2005*), Mongolian gerbils (*Budinger et al., 2000*; *Ohga et al., 2017*; *Thomas et al., 1993*), rats (*Polley et al., 2007*), and mice (*Guo et al., 2012*; *King et al., 2018*). Tonotopy originates in the cochlea and is relayed and preserved throughout all the ascending stages, including the medial geniculate body (MGB) and the AuC (*Jasmin et al., 2019*; *Simon et al., 2009*; *Smith and Wever, 1949*). As one of the most explored pathways in the auditory system, the connections between MGB and AuC play an essential role in the perception of auditory information (*Lee, 2013*; *Ohga et al., 2017*; *Pardi et al., 2020*).

Core fields of the AuC receive their predominant thalamic input from the ventral division of the MGB (MGBv), which confers well-defined frequency tuning arranged into smoothly varying tonotopic gradients (*Andersen et al., 1980*; *Merzenich and Brugge, 1973*; *Winer et al., 2005*). Non-core AuC fields are innervated by non-primary divisions of the MGB and from intracortical inputs originating outside of the AuC (*Jones, 2001*; *Lee and Winer, 2005*; *Reale and Imig, 1980*), which show weak tonotopy and selectivity for processing conspecific communication sounds (*Norman-Haignere et al., 2015*; *Schneider and Woolley, 2013*). Feedforward auditory information is conveyed from the MGBv to the AuC, primarily received in layer 4 (L4) but also extending to other layers (*Constantinople and Bruno, 2013*; *Petreanu et al., 2009*). Subsequently, it is projected corticofugally to downstream targets from either layer 5 (L5) or layer 6 (L6) of the AuC (*Shepherd and Yamawaki, 2021*), including feedback information to MGB (*Happel et al., 2014*; *Happel et al., 2010*; *Homma and Bajo, 2021*; *Kirchgessner et al., 2020*). Several studies have revealed the sophisticated inputs and the ability to perceive complex sounds of CT neurons in A1 (*Clayton et al., 2021*; *Homma et al., 2017*), suggesting the descending information flow in the core fields of the AuC possibly contributes to higher-order feedback processing of diverse auditory inputs.

It is also known that the tonotopic gradient is well preserved across all layers within the A1 (*Guo et al., 2012*; *Montes-Lourido et al., 2021*; *Tischbirek et al., 2019*; *Winkowski and Kanold, 2013*). However, all of these previous discoveries in tonotopic mapping are based on layer specificity. While the laminar structure is closely associated with functional projections in the AuC, these are not precisely identical. Neurons within the same layer can exhibit diversity in their molecular, morphological, physiological, and connectional features (*Triarhou, 2021*; *Yarmohammadi et al., 2014*), and therefore, categorizing cortical neurons based on their projection specificity can provide a deeper fundamental understanding of their functional organization. The A1-MGB projecting neurons play an essential role in the feedforward-feedback loop of the thalamus-neocortex interplay for integrated sound information processing flow (*Happel et al., 2010*) and the perception of complex sounds (*Clayton et al., 2021*; *Homma et al., 2017*). However, due to technical difficulties, there has been limited understanding of the functional properties of these specific corticothalamic projecting neurons (*Antunes and Malmierca, 2021*; *Winer et al., 2001*), because they are located in deeper layers of the cortex and are more difficult to access by means of commonly used, single-cell-resolving recording techniques such as conventional two-photon (2P) microscopy (*Kobat et al., 2009*; *Oheim et al., 2001*; *Takasaki et al., 2020*; *Tischbirek et al., 2015*).

Recent work has established a rabies-based retrograde labeling method with pathway specificity (*Sun et al., 2019*; *Zhu et al., 2020*) to circumvent the neuropil contamination problem associated with conventional 2P imaging for CT neurons. Here, using viral tracing and 2P Ca$^{2+}$ imaging of awake head-fixed mice, we systematically revisited the functional organization in the A1, specifically targeting two types of neurons with projection specificity: TR (*Zingg et al., 2017*) and CT neurons (*Gu et al., 2023*), akin to the input and output of the AuC. The projection specificity-based approach provides detailed insight into the differences in population activity and frequency response-related topographic organization between neurons with different connectivity specificities.

## Results

### Pathway-specific labeling of TR and CT neurons

To label TR neurons, we used an anterograde trans-synaptic tracing strategy by injecting a Cre-recombinase-expressing AAV into the MGB, then injecting a second AAV carrying Cre-dependent GCaMP6s (AAV2/9-CaMKII-DIO-GCaMP6s) in the AuC (*Figure 1A and B*; *Zingg et al., 2017*). Neurons labeled with AAV-GCaMP6s were observed across all layers of the AuC (*Figure 1C and D*). To label CT neurons, we performed retrograde tracing with a rabies virus CVS-N2c-derived vector (CVS-GCaMP6s) by injecting it into the MGB (*Figure 1E*). Neurons labeled with CVS-GCaMP6s were restricted to AuC L6 (*Figure 1F and G*) and were then imaged by 2P Ca$^{2+}$ imaging, in line with our recent work (*Gu et al., 2023*). Whole-cell patch-clamp recordings and in vivo 2P imaging showed no change in electrophysiological, morphological, or functional characteristics of CVS-labeled CT neurons (*Gu et al., 2023*). To locate A1, we conducted wide-field imaging in awake GCaMP6s-labeled mice and defined that the low-frequency (LF) tone (4 kHz) elicited spatially restricted responses in the regions referred to as 'LF area', and that the high-frequency (HF) tone (32 kHz) elicited responses in the regions referred to as 'HF area' (*Figure 1H*). In AAV-GCaMP6s labeled mice, we identified the locations of A1, anterior auditory field (AAF), and secondary auditory field (A2) based on the known tonotopy in mice (*Figure 1H*, left panel) (*Issa et al., 2014*; *Liu et al., 2019*).

It is worth noting that, in CVS-GCaMP6s-labeled (i.e. CT neurons) mice, pure tone stimulation elicited wide-field signals in only one auditory region that exhibited a tonotopic gradient (*Figure 1H*, right panel). To identify this region in AuC, we injected a retrograde tracing indicator cholera toxin subunit B conjugated with Alexa 555 (CTB-555) into this region, which resulted in clearly visible CTB-555-labeled cell bodies in the MGBv (*Figure 1I*), suggesting that CVS-GCaMP6s-labeled neurons were located in A1 (*Rothschild et al., 2010*). When performing 2P imaging, the average power delivered to the brain was in the range of 30—120 mW, depending on the depth of imaging. To determine whether neurons labeled with the CVS retained normal response properties after deep 2P imaging – with higher laser power than that used for superficial layer imaging – we observed the broadband noise-evoked (BBN-evoked) responses of these neurons over time (days) (*Figure 1—figure supplement 1A*). At 7 and 13 days after injection, we found that CVS-labeled neurons had completely normal response properties (*Figure 1—figure supplement 1B and C*).

Next, we performed Ca$^{2+}$ imaging of TR or CT neurons from A1 identified by wide-field imaging (*Figure 1J*). High-titer AAV1-Cre virus has been reported to anterogradely label postsynaptic neurons and also retrogradely label presynaptic neurons (*Zingg et al., 2017*). To minimize this labeling ambiguity, we restricted the 2P imaging depth of TR neurons to 200—400 μm, without imaging the neurons in L5 and L6, which were reciprocally connected with the MGB (*Harris and Mrsic-Flogel, 2013*). These results suggest the feasibility of our approach to label and record population Ca$^{2+}$ signals with pathway specificity at single-cell resolution in awake mice.

### CT neurons exhibit no tonotopic gradient

To study tonal response profiles, we delivered 330 pure tones at each focal plane (11 frequencies ranging from 2 to 40 kHz, 6 attenuation levels, 5 repetitions). We determined the best frequency (BF) (*Guo et al., 2012*) of all tuned TR (*Figure 2A*) or CT neurons (*Figure 2B*) at each 2P imaging focal plane. In total, we imaged 1041 TR neurons in 5 animals. 46% were 'tuned' neurons, 41% were 'irregular' neurons, and 13% were 'silent' neurons (*Figure 2—figure supplement 1A*, the definitions of 'irregular', 'tuned', and 'silent', see Materials and methods). For CT neurons, we imaged a total of 2721 neurons from 10 animals. 18% were 'tuned' neurons, 57% were 'irregular' neurons, and 25% were 'silent' neurons (*Figure 2—figure supplement 1B*).

Based on the BFs, we determined the median BF of all tuned neurons in each 2P focal plane (e.g. *Figure 2C and D*) and related them to the position within the wide-field imaging areas (*Figure 2E and F*), which were then used for computing the tonotopic gradient. For TR neurons, we observed a clear tonotopic axis in the organization of median BF (R=0.77, p<0.001; 2.13 octaves/mm; *Figure 2G*). However, no such tonotopic gradient was observed in CT neurons (R=0.20, p=0.21; 0.21 octaves/mm; *Figure 2H*). Plotting the relative positions of all individual tuned TR or CT neurons in each mouse revealed a significant low-to-high frequency gradient from caudal to rostral for TR neurons (R=0.52, p<0.001; 1.76 octaves/mm; *Figure 2I and J*), but an absence of tonotopy in CT neurons (R=0.079, p=0.084; 0.37 octaves/mm; *Figure 2K and L*). These results, at single-neuron resolution, confirmed

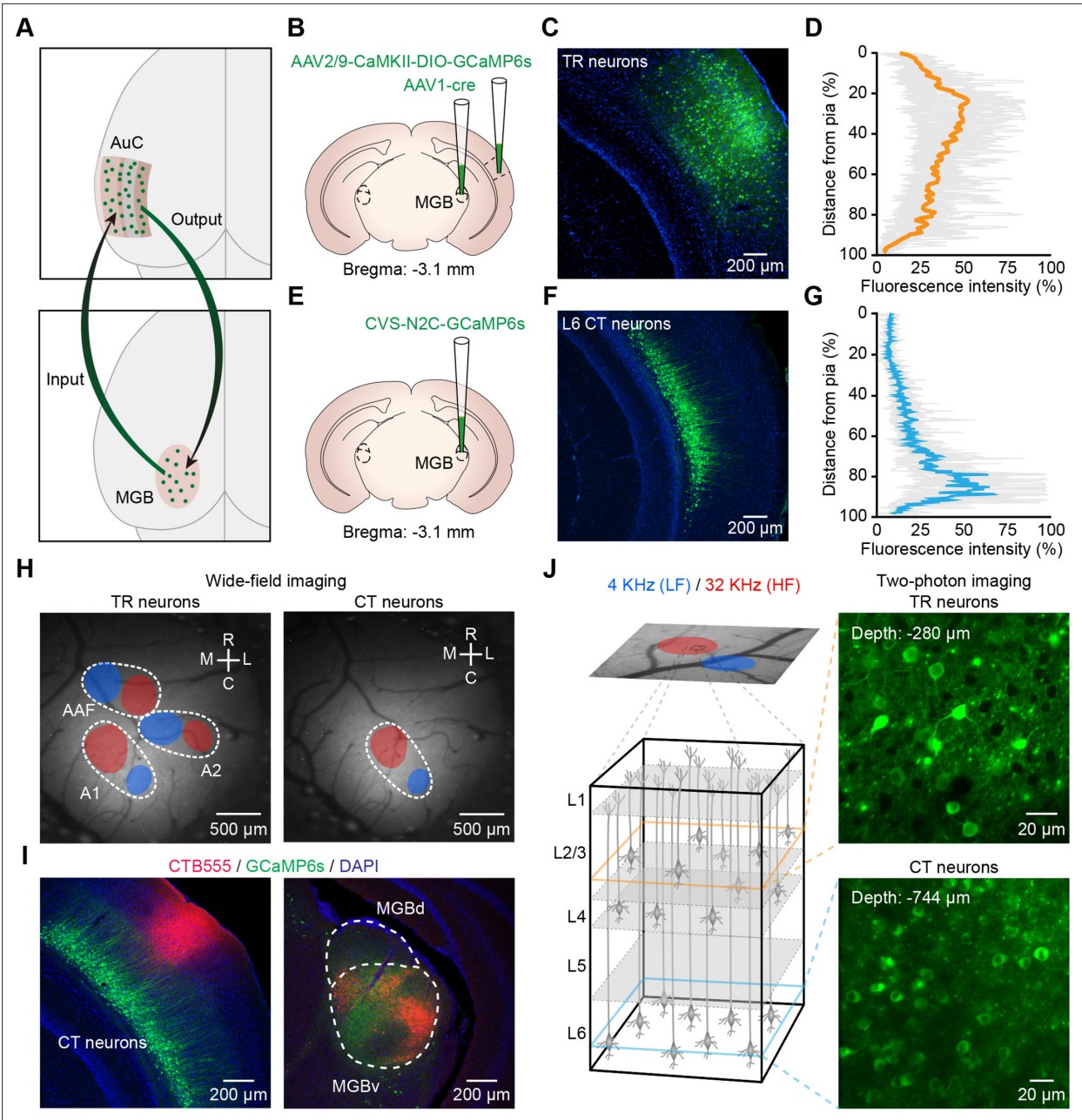

**Figure 1.** Imaging thalamocortical recipient (TR) or corticothalamic (CT) neurons in primary auditory cortex (A1) of awake mice. (**A**) Cartoon illustration of the auditory thalamocortical and corticothalamic circuits. (**B**) Schematic diagram of the injection site of AAV2/1-Cre in the medial geniculate body (MGB) and AAV2/9-CaMKII-DIO-GCaMP6s in auditory cortex (AuC). (**C**) Coronal slice showing AAV-GCaMP6s expression in the AuC on day 21. (**D**) Fluorescence intensity of TR neurons in AuC with distance from pia (0%) to the L6/WM border (100%). (**E**) Schematic diagram of the injection site of CVS-GCaMP6s in the MGB. (**F**) Coronal slice showing the CVS-GCaMP6s retrogradely labeled neurons in the AuC on day 7. (**G**) Fluorescence intensity of CT neurons in AuC with distance from pia (0%) to the L6/WM border (100%). (**H**) Left: wide-field imaging in AAV-GCaMP6s-expressing mouse (TR neurons), fluorescence response to pure tone stimulation with 4 kHz (blue area) and 32 kHz (red area). White dotted lines outline the A1, A2, and AAF boundaries. Right: wide-field imaging in CVS-GCaMP6s-expressing mouse (CT neurons). (**I**) Left: a fluorescent micrograph of a coronal slice after CTB-555 loading guided by two-photon (2P) imaging into A1 in a CVS-GCaMP6s-expressing mouse. Right: micrograph of a coronal slice of the MGB from the same mouse. CTB-555 retrogradely labeled neurons were mainly concentrated in the ventral division of the MGB (MGBv). (**J**) Left: wide-field imaging (in a GCaMP6s-expressing mouse) in A1, fluorescence response to pure tone stimulation with 4 kHz (low-frequency [LF] area) and 32 kHz [HF] area), same abbreviation for all subsequent figures. Inset panels outlined by dashed boxes show the magnified views of 2P imaging of TR or CT neurons. Right: examples of 2P images of TR and CT neurons in vivo, respectively.

The online version of this article includes the following figure supplement(s) for figure 1:

**Figure supplement 1.** Chronic in vivo two-photon (2P) imaging of corticothalamic (CT) neurons.

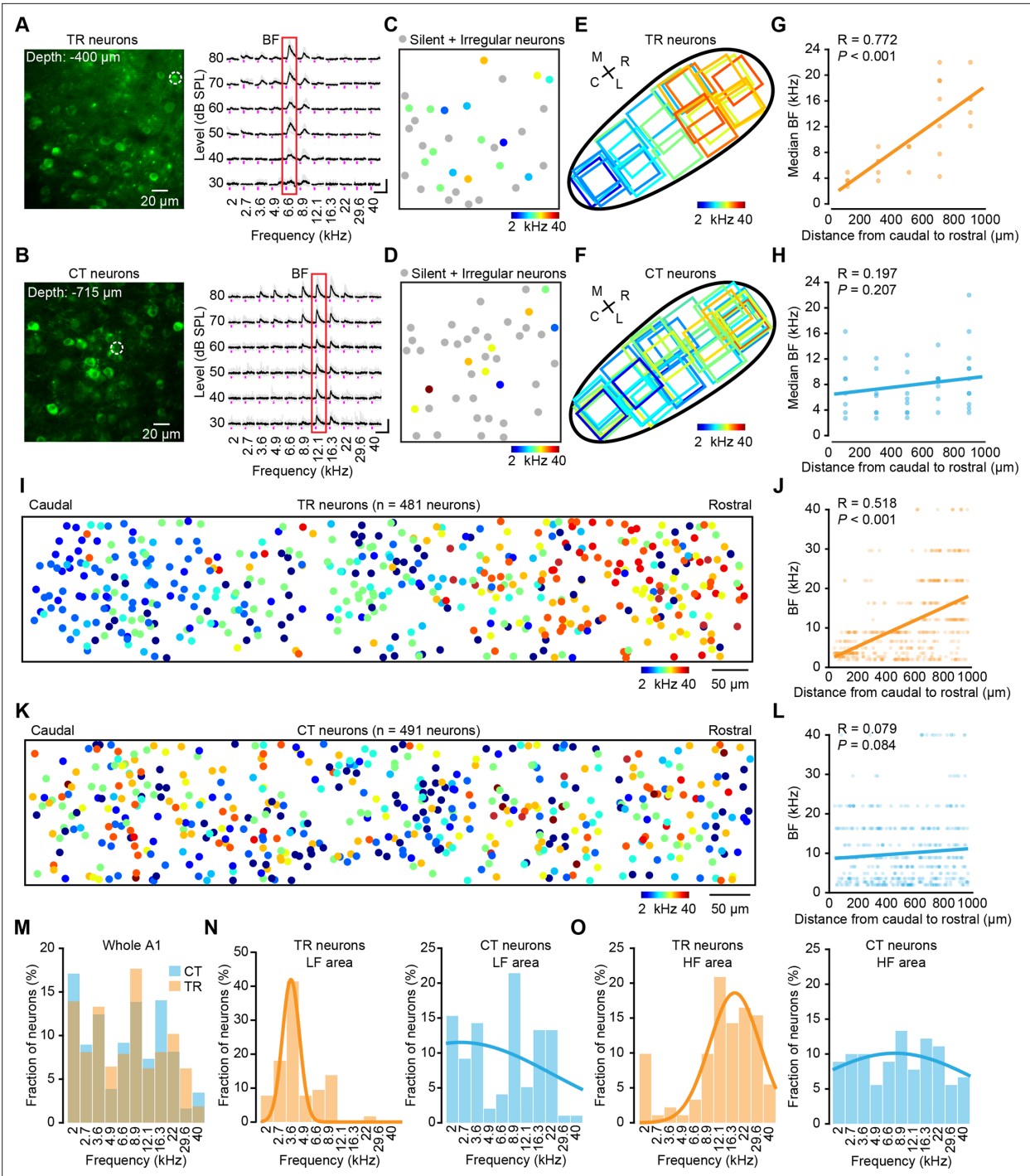

**Figure 2.** Tonotopic gradients of thalamocortical recipient (TR) versus corticothalamic (CT) neurons. (**A**) Two-photon (2P) image of the focal plane used to image TR neurons in primary auditory cortex (A1). The dashed circle indicates the tuned neurons. Fluorescence traces of neurons in the right panel are indicated by the dashed circle ordered according to sound frequency and level. The five traces associated with the five repeats of each stimulus are plotted in gray. The average calcium signals are plotted in black. The red outline marks the best frequency (BF) response of the neuron. (**B**) Same as panel (**A**) but for CT neurons in A1. (**C**) BF map of TR neurons. (**D**) BF map of CT neurons. (**E**) Schematic illustrating the recording locations of individual fields of view of TR neurons in A1. Outlines are color-coded according to the median BF response in the respective fields of view, with their BFs (kHz) color-coded by the scheme below. N=23 focal planes, from 5 mice. (**F**) Same as panel (**E**) but for CT neurons. N=40 focal planes, from 10 mice. (**G**) $BF_{median}$ plotted against distance along the tonotopic axis for TR neurons. Scatter plots showing the correlation between the cellular $BF_{median}$ values measured in the fields of view with 2P imaging and the corresponding extrapolated brain surface frequencies determined by wide-field imaging of TR neurons. (**H**) Same as panel (**G**) but for CT neurons. (**I**) Reconstruction of the relative spatial locations of tuned TR neurons that covered the whole

*Figure 2 continued on next page*

*Figure 2 continued*

A1 area and color-coded according to each neuron's BF. N=481 neurons from 5 mice. (**J**) Plots of TR neurons' BF and their relative distances along the caudal-to-rostral axis. (**K**) Same as panel (I) but for CT neurons. N=491 neurons from 10 mice. (**L**) Same as panel (J) but for CT neurons. (**M**) Distribution histogram of BF from all neurons in the dataset (TR: N=481 neurons; CT: N=491 neurons). (**N**) Distribution histogram of BF neurons that were identified in the LF area by wide-field imaging (TR: N=116 neurons; CT: N=98 neurons). (**O**) Distribution histogram of BF neurons that were identified in the HF area by wide-field imaging (TR: N=91 neurons; CT: N=90 neurons).

The online version of this article includes the following figure supplement(s) for figure 2:

**Figure supplement 1.** Pie charts showing the percentage of responsive neurons to pure tone stimulation of thalamocortical recipient (TR) and corticothalamic (CT) neurons.

**Figure supplement 2.** Tonotopic gradients of general excitatory neurons in superficial layers.

**Figure supplement 3.** Comparison of thalamocortical recipient (TR) neurons across superficial layers.

**Figure supplement 4.** Imaging pure tone responses of non-thalamocortical recipient (TR) neurons.

that the TR neuronal population does possess tonotopy, as previous reports have shown on a coarse scale (*Kalatsky et al., 2005*; *Wu et al., 2006*).

The fraction of different BF-responsive neurons among TR and CT neurons showed that both TR and CT neurons displayed no obvious pattern throughout A1 (*Figure 2M*). When categorized into LF or HF areas, higher proportions of TR neurons responded to LF sounds in the LF areas or HF sounds in the HF areas, whereas CT neurons showed no frequency preference in either area (*Figure 2N and O*). Note the obvious discrepancy in that the 2P imaging data here shows no tonotopic gradient of L6 CT neurons, but wide-field epifluorescence imaging data shows the presence of a tonotopic gradient from the same mice (*Figure 1H*), which could be largely due to differences in the imaging methods and will be discussed later on.

To validate the reliability of our imaging/analysis approach, we performed control experiments that imaged the general excitatory neurons in superficial layers (*Figure 2—figure supplement 2*), and the results showed a clear tonotopic gradient, which was consistent with previous findings (*Bandyopadhyay et al., 2010*; *Romero et al., 2020*; *Rothschild et al., 2010*; *Tischbirek et al., 2019*). Moreover, to definitively disentangle projection-specific properties from general layer-specific properties, we first analyzed TR neurons to see if response properties varied by depth within the superficial layers. We found no significant differences in the fraction of tuned neurons, field interquartile range (IQR), or $BW_{max}$ between TR neurons in L2/3 versus those in L4 (*Figure 2—figure supplement 3*). This suggests a degree of functional homogeneity within the thalamorecipient population across these layers. To directly test if projection identity confers distinct functional properties within the same cortical layers, we performed the crucial control of comparing TR neurons to their neighboring non-TR neurons. Our results show that TR neurons are significantly more likely to be tuned to pure tones than their neighboring non-TR excitatory neurons (*Figure 2—figure supplement 4*). This finding provides direct evidence that a neuron's long-range connectivity, and not just its laminar location, is a key determinant of its response properties.

## CT neurons exhibit high heterogeneity in tuning preference

Next, we performed an analysis of the BF spatial distribution within each focal plane. Examination of BF distribution among TR or CT neurons revealed that TR neurons exhibited similar frequency preference in HF or LF areas (*Figure 3A*), whereas CT neurons displayed high heterogeneity in their frequency preferences, regardless of LF or HF areas (*Figure 3B*).

To quantify the degree of BF heterogeneity, we examined BF spatial distribution at the focal plane, local, and nearest-neighboring scales. To this end, the interquartile range of BF ($IQR_{BF}$, in octaves) was calculated for all tuned neurons in each focal plane (*Figure 3C and D*). Quantitative analysis showed that CT neurons had higher $IQR_{BF}$ than TR neurons across all focal planes (*Figure 3C*). Similarly, evaluation of LF or HF areas indicated that CT neurons had significantly higher $IQR_{BF}$ than TR neurons in LF areas but showed no significant difference from TR neurons in HF areas at the focal plane level (*Figure 3D*).

To assess tone response heterogeneity at the local scale, we then calculated $IQR_{BF}$ for all tuned neurons within a 25 μm radius (*Winkowski and Kanold, 2013*; *Zeng et al., 2019*) around each randomly selected CT or TR neuron (*Figure 3E—G*). The results revealed that TR neurons also exhibited lower

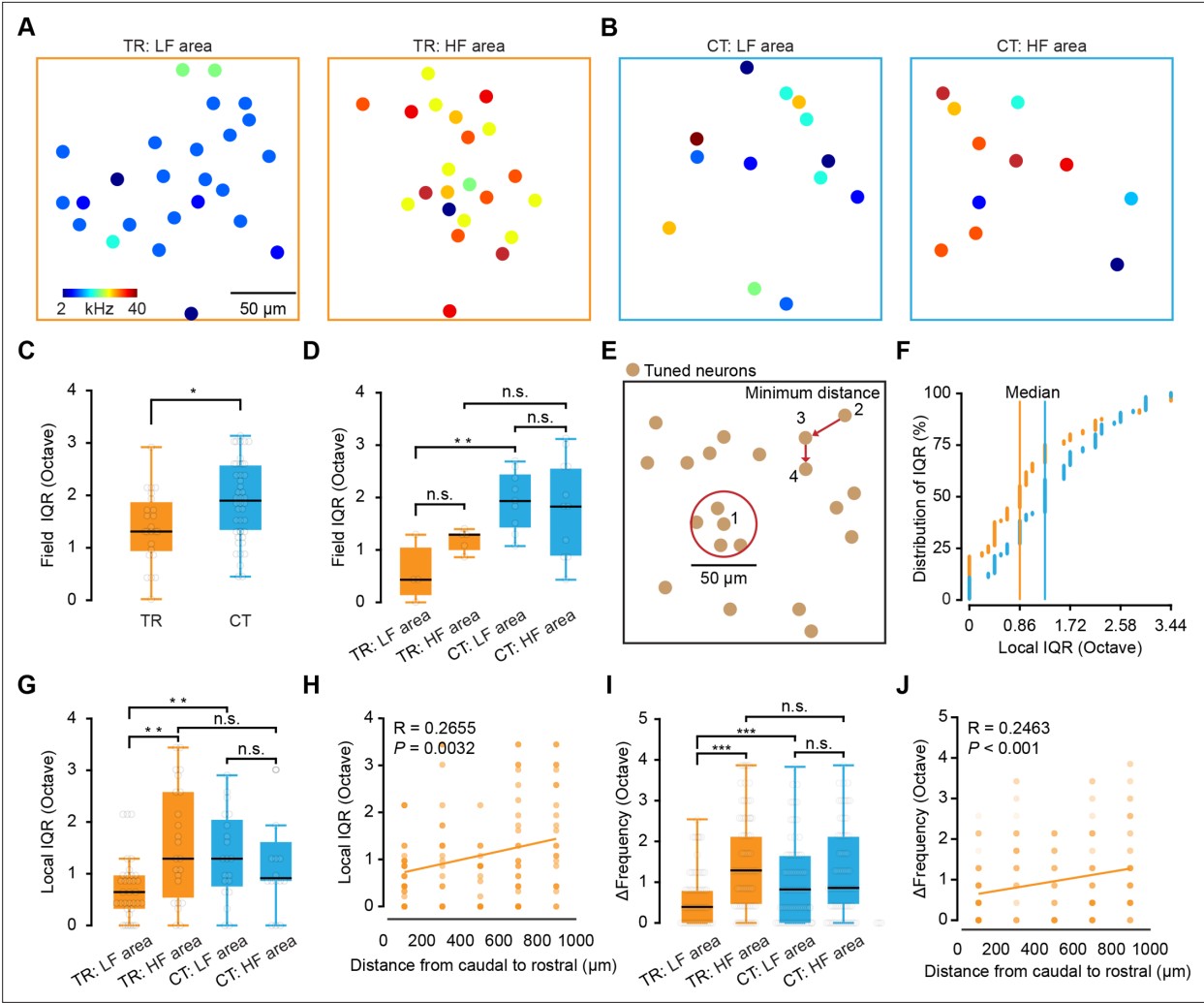

**Figure 3.** Heterogeneity of frequency preference in thalamocortical recipient (TR) versus corticothalamic (CT) neurons. (**A**) Spatial distribution of all tone-selective neurons in an example imaging plane of TR neurons from low-frequency (LF) (left) and high-frequency (HF) (right) imaging areas. (**B**) Same as panel (**A**) but for CT neurons. Best frequencies (BFs) (kHz) are color coded on panel (**A**). (**C**) Comparison of the field (200 μm) interquartile range (IQR) of TR and CT. N=23 focal planes from 5 mice for TR; N=40 focal planes from 10 mice for CT. 'TR': 1.29\0.91–1.88 octaves (median\25–75% percentiles, same notation for all subsequent data), 'CT': 1.94\1.29–2.58 octaves. p=0.016, two-sided Wilcoxon rank-sum test, *p<0.05, **p<0.01, and ***p<0.001, same statistics for boxplots. (**D**) Comparison of the field IQR of TR and CT neurons in LF and HF areas. LF area: N=4 focal planes from 3 mice for TR; N=9 focal planes from 5 mice for CT. HF area: N=5 focal planes from 5 mice for TR; N=11 focal planes from 10 mice for CT. 'TR: LF area': 0.43\0.22–0.86oc taves, 'TR: HF area': 1.29\1.02–1.32oc taves, 'CT: LF area': 1.94\1.45–2.42oc taves, 'CT: HF area': 1.83\0.94–2.58oc taves. p (TR: LF area, HF area)=0.11, p (CT: LF area, HF area)=0.82, p (LF area: TR, CT)=0.0084, p (HF area: TR, CT)=0.25. (**E**) Cartoon showing how the local and minimum percentages were computed. (**F**) Cumulative percentage plot displaying local (25 μm radius) IQR around each neuron of TR and CT. Median local IQR$_{BF}$ of TR neurons = 0.86 octaves, N=120 local planes from 5 mice; median local IQR$_{BF}$ of CT neurons = 1.29 octaves, N=105 local planes from 10 mice. (**G**) Comparison of the local (25 μm radius) IQR of TR and CT neurons in LF and HF areas. LF area: N=34 local planes for TR; N=20 local planes for CT. HF area: N=23 local planes for TR; N=16 local planes for CT. 'TR: LF area': 0.65\0.32–0.97 octaves, 'TR: HF area': 1.29\0.54–2.58 octaves, 'CT: LF area': 1.29\0.75–2.04 octaves, 'CT: HF area': 0.91\0.86–1.61 octaves. p (TR: LF area, HF area)=0.0021, p (CT: LF area, HF area)=0.69, p (LF area: TR, CT)=0.0057, p (HF area: TR, CT)=0.34. (**H**) Local IQR$_{BF}$ plotted against distance along the tonotopic axis for TR neurons. (**I**) Comparison of the minimum distance ΔFrequency of TR and CT neurons in LF and HF area. LF area: N=116 paired neurons for TR; N=87 paired neurons for CT. HF area: N=91 paired neurons for TR; N=78 paired neurons for CT. 'TR: LF area': 0.43\0.00–0.86 octaves, 'TR: HF area': 0.86\0.00–1.72 octaves, 'CT: LF area': 0.43\0.00–1.29 octaves, 'CT: HF area': 0.43 \0.00–1.29 octaves. p (TR: LF area, HF area)=4.25e-04, p (CT: LF area, HF area)=0.79, p (LF area: TR, CT)=0.18, p (HF area: TR, CT)=0.025. (**J**) ΔFrequency plotted against distance along the tonotopic axis for TR neurons.

median $IQR_{BF}$ than CT neurons at the local scale (*Figure 3F*). Consistent with the focal plane scale, CT neurons had significantly higher local $IQR_{BF}$ than TR neurons in LF areas (*Figure 3G*). Unexpectedly, the local $IQR_{BF}$ of TR neurons in HF areas was significantly higher than that of TR neurons in LF areas (*Figure 3G*). Further analysis across local regions revealed that the local $IQR_{BF}$ of TR neurons followed a low-to-high gradient along the caudal-to-rostral axis (R=0.2655, p=0.0032; *Figure 3H*).

The analysis of Δfrequency (BF variability) (*Bandyopadhyay et al., 2010*; *Zeng et al., 2019*) between nearest-neighboring TR or CT neurons was consistent with results of local $IQR_{BF}$ analysis (*Figure 3I*). That is, Δfrequency followed a clear gradient in the organization of TR neurons along the caudal-to-rostral axis (R=0.2463, p<0.001; *Figure 3J*). Moreover, TR neurons, but not CT neurons, displayed high $IQR_{BF}$ in HF areas compared to LF areas (*Figure 3G and I*). These results suggest the relative heterogeneity of frequency preference among CT neurons but homogeneity among TR neurons.

## Receptive field properties of TR versus CT neurons

Given the above differences in the functional organization of frequency preference between TR and CT neurons, we next focused our analysis on the frequency responsive areas (FRAs) of individual neurons, as the FRAs directly reflect neuronal auditory selectivity essential for sound processing (*Sadagopan and Wang, 2008*). For this analysis, all tuned neurons were categorized into either a V-shaped (decreasing frequency selectivity with increasing intensity), I-shaped (narrow, level-tolerant tuning), or O-shaped (non-monotonic) FRA (*Figure 4A and B*). We found that a higher proportion of CT neurons was associated with V-shaped FRAs than TR neurons, whereas the proportion of CT neurons associated with I-shaped FRAs was lower than that of TR neurons (*Figure 4C*). By contrast, no difference was detected in the proportions of CT and TR neurons associated with O-shaped FRAs (*Figure 4C*).

A key feature of neuronal tuning curves is their sharpness. Common measures of sharpness in V shape are half-peak bandwidth and the 'quality factor' (Q), which was obtained by dividing the BF of the neuron by a measure of tuning (*Micheyl et al., 2013*), the width of the tuning curve at half-peak in this study. This analysis showed that the half-peak bandwidth of CT neurons was significantly wider than that of TR neurons (*Figure 4D*), while the Q value of CT neurons was lower than that of TR neurons, especially in the HF areas (*Figure 4E and F*). Since higher Q values indicate sharper tuning of neurons, which suggests higher frequency discrimination (*Micheyl et al., 2013*), these results suggest that TR neurons might be responsible for finer tone discrimination.

As the bandwidth of neurons' FRA can also reflect the selectivity of overall tonal responses (*Rodrigues-Dagaeff et al., 1989*; *Schreiner and Sutter, 1992*), we examined the maximum FRA width of TR or CT neurons (*Figure 4G–J*). $BW_{max}$ was defined as the maximum FRA width at any sound level. The quantitative analysis showed that the $BW_{max}$ of TR neurons was significantly narrower than that of CT neurons, irrespective of the FRA shape (*Figure 4G*). In addition, the distribution of TR neurons generally peaked at a $BW_{max}$ of 1—2 octaves, while the distribution of CT neurons peaked at a $BW_{max}$ of 3—4 octaves (*Figure 4H*). The statistical analysis showed that CT neurons had significantly higher $BW_{max}$ than TR neurons (*Figure 4I*). Within LF or HF areas, TR neurons had a significantly narrower $BW_{max}$ range than CT neurons in either area. Surprisingly, TR neurons in HF areas had a significantly lower $BW_{max}$ than TR neurons in LF areas (*Figure 4J*), suggesting that the $BW_{max}$ distribution among TR neurons might change along the caudal-to-rostral axis.

We then investigated the organization for intensity tuning, another crucial feature in auditory perception, based on the monotonicity index (MI, see Materials and methods for details), wherein 1 indicated monotonic response increase with intensity and 0 indicated strong nonmonotonic tuning (*de la Rocha et al., 2008*; *Sutter and Schreiner, 1995*; *Watkins and Barbour, 2011*). The MI distribution across both TR and CT neuronal populations peaked at 1 (*Figure 4K*). Subsequent categorization as either monotonic or non-monotonic (using a criterion of MI = 0.5 [*de la Rocha et al., 2008*; *Moore and Wehr, 2013*], neurons with an index <0.5 considered intensity-tuned) showed that the fraction of neurons exhibiting non-monotonic intensity response did not significantly differ between TR and CT neurons (*Figure 4L*). However, examination of MI values within LF or HF areas showed that a higher proportion of non-monotonic CT neurons than TR were located in LF areas, while a larger proportion of non-monotonic TR neurons was located in HF areas than in LF areas (*Figure 4M*). In addition, among non-monotonic neurons, TR neurons had higher MI than CT neurons, whereas monotonic TR neurons had a lower MI than monotonic CT neurons (*Figure 4N*). Furthermore, we note that

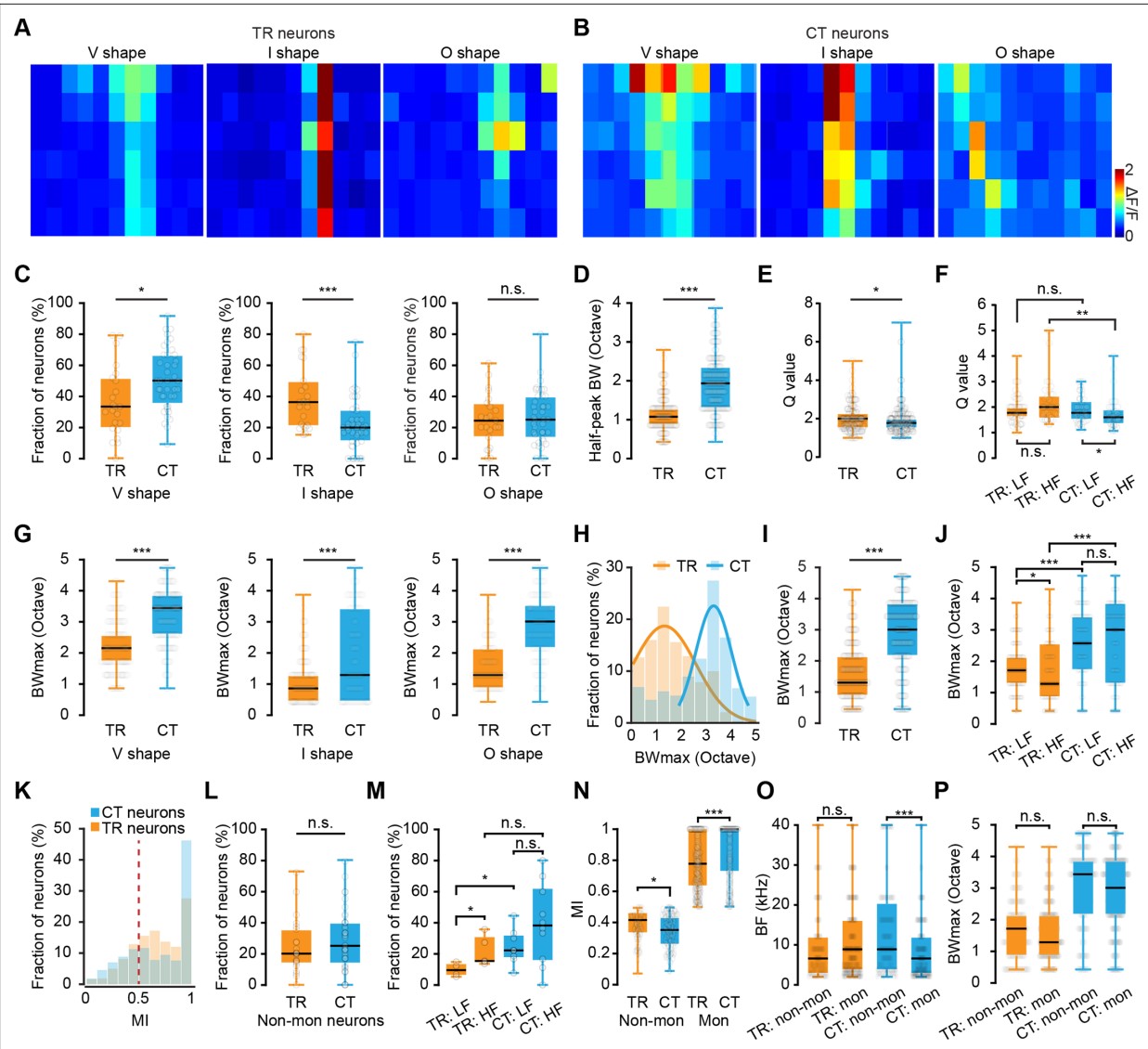

**Figure 4.** Receptive field properties of thalamocortical recipient (TR) versus corticothalamic (CT) neurons. (**A**) One representative frequency responsive area (FRA) from a specific type of TR neuron. Types are V shape, I shape, and O shape. See text for further details. The X-axis represents 11 pure tone frequencies, and the Y-axis represents six sound intensities. (**B**) One representative FRA from a specific type of CT neuron. (**C**) Comparison of fraction of neurons from V-, I-, and O-shaped neurons in TR and CT. (**D**) Comparison of half-peak bandwidth of V-shaped TR and CT neurons. N=190 neurons from 5 mice for TR; N=249 neurons from 10 mice for CT. 'TR': 1.08\0.86–1.29 octaves, 'CT': 1.94\1.29–2.37 octaves. p=4.86e-29. (**E**) Comparison of Q value (bandwidth of best frequency [BF]/half bandwidth) of V-shaped TR and CT neurons. N=190 neurons from 5 mice for TR; N=249 neurons from 10 mice for CT. 'TR': 2.00\1.50–2.29, 'CT': 1.78\1.50–2.00. p=0.035. (**F**) Comparison of the Q value of V-shaped TR and CT neurons in LF and HF areas. LF area: N=56 neurons for TR; N=49 neurons for CT. HF area: N=37 neurons for TR; N=41 neurons for CT. 'TR: LF area': 1.78\1.60–2.00, 'TR: HF area': 2.00\1.58–2.43, 'CT: LF area': 1.78\1.50–2.22, 'CT: HF area': 1.60\1.33–1.88. p (TR: LF area, HF area)=0.20, p (CT: LF area, HF area)=0.032, p (LF area: TR, CT)=0.88, p (HF area: TR, CT)=0.0032. (**G**) Comparison of maximum bandwidth (BW$_{max}$) from V-, I-, and O-shaped neurons in TR and CT. V shape: N=190 neurons for TR, N=249 neurons for CT; 'TR': 2.15\1.72–2.58 octaves, 'CT': 3.44\2.58–3.87 octaves; p=4.18e-29. I shape: N=174 neurons for TR, N=112 neurons for CT; 'TR': 0.86\0.43–1.29 octaves, 'CT': 1.29\0.43–3.44 octaves; p=4.66e-05. O shape: N=117 neurons for TR, N=130 neurons for CT; 'TR': 1.29\0.86–2.15 octaves, 'CT': 3.01\2.15–3.44 octaves; p=4.30e-19. (**H**) Distribution histogram of BW$_{max}$ from all neurons in the dataset. (**I**) Comparison of the BW$_{max}$ of TR and CT neurons. N=481 neurons from 5 mice for TR, N=491 neurons from 10 mice for CT; 'TR': 1.29\0.86–2.15 octaves, 'CT': 3.01\2.15–3.87 octaves; p=1.89e-51. (**J**) Comparison of the BW$_{max}$ of TR and CT neurons in LF and HF areas. LF area: N=116 neurons for TR; N=98 neurons for CT. HF area: N=91 neurons for TR; N=90 neurons for CT. 'TR: LF area': 1.72\1.29–2.15, 'TR: HF area': 1.29\0.86–2.47, 'CT: LF area': 2.58\1.72–3.44, 'CT: HF area': 3.01\1.29–3.87. p (TR: LF area, HF area)=0.034, p (CT: LF area, HF area)=0.92, p (LF area: TR, CT)=2.44e-08, p (HF area: TR, CT)=2.28e-06. (**K**) Distribution of the monotonicity index (MI) of TR and CT neurons. (**L**) Comparison of the fraction of non-monotonic neurons of TR and CT. N=23 focal planes from 5 mice for TR, N=40 focal planes from 10 mice for CT; 'TR': 20.00%\12.92–31.49%, 'CT': 25.00%\14.58–40.00%; p=0.33. (**M**) Comparison of non-monotonic neurons of TR and CT in LF and HF areas. LF area: N=4 focal planes from 3 mice for TR; N=9 focal planes from 5 mice for CT. HF area:

*Figure 4 continued on next page*

*Figure 4 continued*

N=5 focal planes from 5 mice for TR; N=11 focal planes from 10 mice for CT. 'TR: LF area': 9.55%\6.08–13.50%, 'TR: HF area': 15.38%\14.66–29.38%; 'CT: LF area': 22.22%\17.59–31.67%, 'CT: HF area': 38.18%\16.67–60.00%; p (TR: LF area, HF area)=0.032, p (CT: LF area, HF area)=0.19, p (LF area: TR, CT)=0.016, p (HF area: TR, CT)=0.17. (**N**) Comparison of the MI of non-monotonic and monotonic neurons in TR and CT. Non-mon: N=107 neurons for TR, N=128 neurons for CT; 'TR': 0.42\0.33–0.47, 'CT': 0.35\0.26–0.44; p=7.03e-04. Mon: N=374 neurons for TR, N=363 neurons for CT; 'TR': 0.78\0.63–1.00, 'CT': 1.00\0.73–1.00; p=6.19e-11. (**O**) Comparison of the BF between monotonic and non-monotonic neurons in TR and CT. TR: N=107 neurons for non-mon, N=374 neurons for mon; 'non-mon': 6.60\2.93–12.10 kHz, 'mon': 8.90\3.60–16.30 kHz; p=0.69. CT: N=128 neurons for non-mon, N=363 neurons for mon; 'non-mon': 8.90\3.60–19.15 kHz, 'mon': 6.60\2.70–12.10 kHz; p=1.64e-04. (**P**) Comparison of the BW$_{max}$ between monotonic and non-monotonic neurons in TR and CT. TR: N=107 neurons for non-mon, N=374 neurons for mon; 'non-mon': 1.72\0.86–2.15 octaves, 'mon': 1.29\0.86–2.15 octaves; p=0.92. CT: N=128 neurons for non-mon, N=363 neurons for mon; 'non-mon': 3.44\2.15–3.87 octaves, 'mon': 3.01\2.15–3.87 octaves; p=0.052.

non-monotonic CT neurons had a higher BF than monotonic CT neurons, whereas the BF was similar between monotonic and non-monotonic TR neurons (*Figure 4O*). Finally, BW$_{max}$ did not significantly differ between monotonic and non-monotonic neurons in either the TR or CT populations (*Figure 4P*).

## Discussion

A key finding in this study is that the information flow descending from A1 to the thalamus via CT neurons does not preserve tonotopy, which contrasted with the observation that the information flow ascending from the thalamus to A1 via TR neurons exhibits clear tonotopy.

### Layer-specific versus projection-specific functional organization in A1

Previous studies have reported the existence of tonotopic gradients across all layers of A1 including L6 (*Guo et al., 2012*; *Tischbirek et al., 2019*). CT neurons labeled with CVS-GCaMP6s were restricted to A1 L6 but did not preserve tonotopy. This is because only 30–50% of the pyramidal cells in L6 are CT neurons (*Thomson, 2010*), and the neurons we image are restricted to MGB-projecting CT neurons, excluding CT neurons projecting to other downstream nuclei (*Clayton et al., 2021*). Using a CVS-based labeling approach and 2P single-cell-resolved imaging, we successfully dissected a small fraction of CT neurons from all L6 neurons and studied the functional organization of this group of neurons.

The studies examining functional organization in AuC have mostly focused on layer specificity (*Bandyopadhyay et al., 2010*; *Montes-Lourido et al., 2021*; *Rothschild et al., 2010*; *Tischbirek et al., 2019*; *Winkowski and Kanold, 2013*), and these studies have provided us with a basic understanding of the tonotopy and heterogeneity of AuC. However, layer-specific representation is a relatively coarse approach. A projection specificity-based approach provides detailed insight into differences in population activity and frequency response-related topographic organization between neurons with different connectivity specificity. For example, anatomical studies of AuC in several species have identified the topographical organization of L5 corticocollicular projections (*Bajo and Moore, 2005*; *Saldana and Feliciano, 1996*; *Stebbings et al., 2014*). By contrast, a recent study using a pathway-specific labeling method showed that L5 corticocollicular neurons displayed a relatively weaker topological organization than the non-corticocollicular neurons in the same layer (*Schmitt et al., 2023*). Those studies and our data consistently support the notion that different anatomically organized projections do not necessarily transmit information with the same topological alignment. Therefore, combining pathway-specific labeling and in vivo single-cell-resolved functional imaging could reveal unexpected fine-scale details and settle discrepancies arising from results obtained by individual methods alone.

### CVS virus provides an effective means of mapping neuronal function in L6

The CVSN2c-ΔG rabies virus strain, a recently engineered self-inactivating ΔG rabies virus lacking the polymerase gene, exhibits strong neurotropism and reduced cytotoxicity (*Reardon et al., 2016*). Previous studies have revealed that reliable neural activity using the CVS-GCaMP6s virus can be maintained for at least 21 days, as recorded by fiber photometry (*Lin et al., 2023*). In acute forebrain slices, CVS-N2c-ΔG-hChR2-YFP reliably elicits action potentials in transfected cortical neurons for at least 28 days after infection (*Reardon et al., 2016*). Our recent studies also confirm that CVS-labeled CT neurons exhibit electrophysiological properties that are indistinguishable from those of normal

neurons (*Gu et al., 2023*). Here, chronic 2P calcium imaging shows that CT neurons retain stable sound response properties for at least 13 days (*Figure 1—figure supplement 1*). All of these studies suggest that the CVS virus-labeling method provides an effective means of mapping neuronal circuitry and manipulating neuronal activity in vivo in the mammalian central nervous system.

## Tonotopy of CT neurons can be observed in wide-field but not 2P imaging

Our 2P imaging results are not contradictory to our wide-field epifluorescence imaging data (*Figure 1*), showing a tonotopy in A1 from the same mice that were later used for 2P imaging experiments. First, the source of the fluorescence signal differs between wide-field imaging and 2P imaging. Wide-field epifluorescence imaging reports all fluorescent proteins from cell bodies and neuropils across layers, whereas 2P imaging only examines fluorescent signals from the cell bodies (*Scott et al., 2018*). Furthermore, BF tuning from individual neurons results in a significantly closer match to wide-field mapping before removal of the neuropil contribution, thus confirming that the neuropil produces greater local homogeneity in BFs along with a clearer global tonotopic organization (*Romero et al., 2020*). For L6 CT neurons, the contribution of the neuropil (dendrites) is important due to the specificity of their morphological structure: the cell bodies are located in L6, while their corresponding apical dendrites terminate in L4 (32%) or extend to L1 (68%) (*Olsen et al., 2012*).

## Profiling of auditory features of 'in-and-out' neurons in A1

At the single-neuron level, the tonotopic organization or functional distribution of adjacent neuron clusters in the AuC can be highly heterogeneous in mice (*Issa et al., 2014*; *Tao et al., 2017*; *Winkowski and Kanold, 2013*). Our findings in this study reveal that TR neurons exhibit more homogeneous functional and spatial distribution (*Figure 3*), which contrasts with some other single-cell-resolved studies in the mouse AuC (*Li et al., 2017*; *Rothschild et al., 2010*; *Winkowski and Kanold, 2013*). This discrepancy can also be explained by the pathway specificity, i.e., only a small fraction of neurons in the upper layers of AuC are thalamocortical recipients. It is expected that TR neurons possess high-gradient tonotopy and sharp tuning because feedforward thalamocortical projections are known to possess the same features (*Guo et al., 2012*; *Kanold et al., 2014*; *Winkowski and Kanold, 2013*). What is unexpected is that the same predictions based on the classical literature of functional mapping of AuC (*Guo et al., 2012*; *Imig et al., 1977*; *Recanzone and Schreiner, 1999*; *Rothschild et al., 2010*) do not apply to L6 CT neurons. This fits perfectly with a previous study, which found a transition from precise, homogenous frequency organization in L4 to coarse, diffuse organization in L2/3, suggesting that information flow exhibits diverse selectivity after cortical processing (*King et al., 2018*; *Winkowski and Kanold, 2013*).

What could be the functional meaning of this change in, or elimination of, tonotopic organization? We suggest that the descending information flow of A1-MGB projection could potentially support higher cortical functions such as higher-level feedback processing of complex (*Homma et al., 2017*; *Malmierca et al., 2015*) and behaviorally meaningful sound (*Jeschke et al., 2021*; *Ohl et al., 2001*; *Wang et al., 2020*; *Wang et al., 2022*). In addition to frequency preference, this study also provides evidence illustrating the effects of cortical processing on response properties. For example, $BW_{max}$, which reflects the degree of integration of synaptic inputs from presynaptic neurons (*Kratz and Manis, 2015*; *Li et al., 2019*; *Schreiner and Sutter, 1992*), is 3.0 octaves in CT neurons, but only 1.3 octaves in TR neurons (*Figure 4I*), indicative of an increase in synaptic integration during cortical processing (*Li et al., 2019*).

We propose that the lack of tonotopy is an active computation, not a passive degradation. CT neurons likely pool inputs from a wide range of upstream neurons with diverse frequency preferences. This broad synaptic integration, reflected in their wider tuning bandwidth, would actively erase the fine-grained frequency map in favor of creating a different kind of representation (*Brewer and Barton, 2016*). This transformation away from a classic sensory map may be critical for the function of corticothalamic feedback. Instead of relaying 'what' frequency was heard, the descending signal from CT neurons may convey more abstract, higher-order information – such as the behavioral relevance of a sound, predictions about upcoming sounds, or motor-related efference copy signals – that are not inherently frequency-specific (*Wang et al., 2020*; *Wang et al., 2022*). The descending A1-to-MGB pathway is often considered to be modulatory, shaping thalamic responses rather than driving them

directly. A modulatory signal designed to globally adjust thalamic gain or selectivity may not require, and may even be hindered by, a fine-grained topographical organization.

## Conclusion

In summary, our findings fill in a knowledge gap in the auditory physiology: whereas the corticothalamic feedback projection is known to contribute to higher cognitive processing of behaviorally relevant complex sounds, the basic pure tone map that underlies and facilitates this advanced processing remained unclear. Our results reveal that the sensory information flow descending from the A1 to the thalamus via L6 CT neurons does not arrange tonotopically. The new shift, categorizing cortical neurons based on their projection specificity, represents an advance in the conceptual framework for functional organization.

# Materials and methods

### Key resources table

| Reagent type (species) or resource | Designation | Source or reference | Identifiers | Additional information |
|---|---|---|---|---|
| Genetic reagent (*Mus musculus*) | Mouse: C57BL/6J | Beijing HFK | RRID:IMSR_JAX:000664 | https://www.jax.org/ |
| Recombinant DNA reagent | AAV2/1-hSyn-Cre-WPRE-hGH polyA | BrainVTA | PT-0136 | |
| Recombinant DNA reagent | AAV2/9-CaMKII-DIO-GCaMP6s-WPRE-hGH pA | BrainVTA | PT-0071 | |
| Recombinant DNA reagent | CVS-N2c-ΔG-GCaM6s | BrainCase Co., Ltd. | BC-RV-CVS715 | |
| Peptide, recombinant protein | Alexa Fluor 555-conjugated Cholera Toxin Subunit B | Invitrogen | C34776 | |
| Chemical compound | DAPI | Sigma-Aldrich | Cat# D9542 | |
| Software, algorithm | MATLAB | MathWorks | RRID:SCR_001622 | |
| Software, algorithm | LabVIEW | National Instruments | RRID:SCR_014325 | |
| Software, algorithm | Igor Pro | WaveMetrics | RRID:SCR_000325 | |
| Software, algorithm | Prism | GraphPad | RRID:SCR_002798 | |
| Other | UHU Super Glue | DETAI | N/A | |
| Other | Sun Medical Super Bond C & B Kit Bonding Kit | Super-Bond | N/A | |

### Animals

C57BL/6J male mice (2–3 months of age) were provided by the Laboratory Animal Center at the Third Military Medical University. The mice were housed in a temperature- and humidity-controlled room on a cycle of 12 hr light/dark (lights off at 19:00) with free access to food and water. All animal procedures were approved by the Animal Care Committee of the Third Military Medical University and were performed in accordance with the principles outlined in the National Institutes of Health Guide for the Care and Use of Laboratory Animals.

### Virus injections and confocal imaging

Mice were anesthetized with 1–2% isoflurane in pure oxygen and placed in a stereotactic frame (Beijing Zhongshi Dichuang Technology Development Co., Ltd.). A warm heating pad was used to keep the animals at a proper body temperature (36.5–37.5℃). To achieve TR neurons labeling, a Cre-expressing AAV (AAV2/1-hSyn-Cre-WPRE-hGH polyA, titer ≥1E+13 vg/ml, PT-0136, BrainVTA) was injected into the MGB (AP –3.1 mm, ML 2.0 mm, DV –2.8 mm from dura) and DIO-expressing AAV (AAV2/9-CaMKII-DIO-GCaMp6s-WPRE-hGH pA, PT-0071, BrainVTA) was injected into the AuC (AP –3.1 mm, ML 3.8 mm, DV –1.4 mm from dura with manipulator tilted 25° from the z-axis). For CT neuron labeling, the CVS virus (CVS-N2c-ΔG-GCaM6s), packaged by BrainCase Co., Ltd., Shenzhen, China, was injected into the MGB. In all the injections, approximately 100 nL of virus was performed.

After injection, the pipette was held in place for 10 min before retraction. Tissue glue (3M Animal Care Products, Vetbond) was used for bonding the scalp incision.

Mice were killed for histology 4 weeks after AAV injection or 7 days after CVS injection. For slice preparation, mice were perfused transcardially with 4% paraformaldehyde in phosphate-buffered saline. Brains were postfixed in 4% paraformaldehyde overnight at 4°C and cut into 50 μm sections on a cryostat microtome (Thermo Fisher, NX50, Waltham, MA, USA). Mounted sections were imaged on a scanning confocal microscope (TCS SP5, Leica).

## Auditory stimulation

Sound stimuli were delivered by an ED1 electrostatic speaker driver and a free-field ES1 speaker (both from Tucker Davis Technologies). During experiments, the speaker was put at a distance of ~6 cm to the left ear of the animal. The sound stimulus was produced by a custom-written, LabVIEW-based program (LabVIEW 2012, National Instruments) and transformed to analogue voltage through a PCI6731 card (National Instruments). All the data were obtained at 1 MHz via a data acquisition device (USB-6361, National Instruments) and examined by our custom-made LabVIEW program. For BBN (BBN, bandwidth 0–50 kHz), the sound level was ~65 dB sound pressure level (SPL). We generated a waveform segment of BBN and used the same waveform segment for all experiments involving BBN.

For functional mapping of TR and CT neurons, sequences of randomly ordered pure tones with 11 frequencies (2–40 kHz) at 6 attenuation levels (30–80 dB SPL) were used. Each frequency-attenuation combination was presented five to eight times. Background noise generated by the recording hardware, especially the resonance scanner of the 2P microscope, was measured to be below <30 dB SPL for the relevant frequency range. The use of a specialized soundproof enclosure with wedge-shaped acoustic foam was implemented to significantly reduce external noise interference. These strategies ensured that auditory stimuli were delivered under highly controlled, low-noise conditions, thereby enhancing the reliability and accuracy of the neural response measurements obtained throughout the study. As described in our previous reports, low frequencies (<1 kHz) were major components of background noise. With a spectral density of ~33 dB/sqrt (Hz), the peak of background noise was below 1 kHz. Neither visible light nor other sensory stimuli were present. The duration of a sound stimulus (tone or BBN) was 100 ms.

## In vivo wide-field epifluorescence imaging

A homemade binocular microscope (BM01, SIBET, CAS) with a 4X, 0.2 NA objective (Olympus) was used to record wide-field fluorescence images in the mouse cortex for establishing the reference cortical map of A1. A light-emitting diode (470 nm, M470L4, Thorlabs) was used for blue illumination. Green fluorescence passed through a filter cube was measured at 10 Hz with an sCMOS camera (Zyla 4.2, Andor Technology). Mice injected with AAV-GCaM6s or CVS-GCaM6s were used to functionally identify the region of A1. The mouse was anesthetized by isoflurane and kept on a warm plate (37.5°C). A piece of bone (~5 mm × 5 mm) was removed and replaced by a coverslip (3 mm in diameter). To localize A1, two pure tones (4 and 32 kHz) were repeatedly presented 20 times at an interval of 6 s.

We observed that pure tone stimulation evoked wide-field signals in only one auditory region in CVS-GCaM6s mice. To confirm the region, we used the criterion that the ventral part of the medial geniculate body (MGBv) is connecting with A1. Injections were performed under visual guidance using 2P excitation. In the experiment, we inserted the electrode into the cortical region at a depth of ~500 μm below the surface. We used Alexa Fluor 555-conjugated cholera toxin subunit B (CTB) as the neural tracer and injected the fluorescent CTB solution with 0.5% in phosphate buffer by pressure (700 mbar) for 15 min. Seven days after the fluorescent CTB injection, the mice were anesthetized with pentobarbital (1.0 g/kg ip). The brain was first dissected out and then it was immersed with 4% paraformaldehyde overnight. To visualize fluorescent tracers, a consecutive series of coronal or horizontal sections (50 μm thick) were prepared using a sliding cryotome, and then all sections were mounted onto glass slides and imaged with a stereoscope (Olympus).

## 2P Ca$^{2+}$ imaging in A1

For 2P imaging in head-fixed awake mice, we removed the skin and muscles over the right A1 after local lidocaine injection under isoflurane anesthesia (1–2%). A custom-made plastic chamber (head

post) designed for head-fixed mouse experiments was then glued to the skull with cyanoacrylic glue (UHU). After recovery from surgery for 3 days, the mouse underwent head-fixation training for 3–5 days (from 1 to 4 hr per day). After head-fixation training, mice gradually adapted to this posture and were able to sit comfortably for 4 hr. On the recording day, a small craniotomy was performed under local anesthesia. Then, the field of interest was determined by comparing the wide-field map with the blood vessel patterns to ensure A1 was imaged.

2P imaging was performed with a custom-built 2P microscope system based on a 12.0 kHz resonant scanner (model 'LotosScan 1.0', Suzhou Institute of Biomedical Engineering and Technology). 2P excitation light was delivered by a mode-locked Ti: Sa laser (model 'Mai-Tai DeepSee', Spectra Physics), and a 40×/0.8 numerical aperture (NA) water immersion objective (Nikon) was used for imaging. For $Ca^{2+}$ imaging experiments, the excitation wavelength was set to 920 nm. The typical size of the field of view (FOV) was ~200 μm × 200 μm. The average power delivered to the brain was in the range of 30–120 mW, depending on the depth of imaging.

## Chronic 2P $Ca^{2+}$ imaging in L6

For chronic 2P imaging, a circular cranial window (2.5 mm diameter coverslip) was implanted over the right AuC after head-post implantation. To this end, the muscle and skull fascia above the temporal skull were removed after local lidocaine (2%) injection. Afterward, a custom-made plastic chamber was fixed to the skull with dental cement (Superbond, Sun Medical Co., Ltd.) and a small craniotomy (~2.7 mm in diameter, the center point: AP: –3.0 mm, ML: –4.5 mm) was performed. The dura was removed, and the craniotomy was polished to match the size of the coverslip. A coverslip was carefully placed on top of the cortex with mild compression by tweezers. The coverslip was sealed with UV-curing dental cement (Tetric N-flow, Ivoclar Co., Ltd.). Antibiotics (Cefazolin, 500 mg/kg, North China Pharmaceutical Group Corporation) were administered before surgery, as well as until 3 days after surgery. 3 days of post-surgery recovery were needed before head-fixation training.

Continuous 2P $Ca^{2+}$ imaging was performed on days 7 and 13. To minimize the bleaching of cells, we used low laser illumination in the deep imaging, based on high-quality imaging of CT neurons with CVS labeling strategy. The repeated imaging FOVs were identified on consecutive days based on superficial blood vessels and nearby blood vessels, then further refined by visually matching reference images acquired from precedent days.

## Quantification and statistical analysis

Data were analyzed using custom-written software in LabVIEW 2012 (National Instruments), Igor Pro 5.0 (WaveMetrics), Image 1.51 (NIH), Prism 8.4 (GraphPad), and MATLAB 2014a (MathWorks).

## Wide-field imaging data analysis

In each mouse, the recorded cortical images were first downsampled from the original 750×1200 pixels to 75×120 pixels. After that, the frames recorded with sound stimuli were averaged across 20 trials. To enhance the signal-to-noise ratio, spatial averaging was conducted over 5×5 pixels by a matrix filter, and temporal averaging was conducted with three consecutive images. The pre-processed images were then temporally normalized to obtain the relative changes in fluorescence (f) pixel-by-pixel. With the baseline fluorescence (f0) obtained by averaging the images of 800 ms before sound stimulation, the relative fluorescence changes of each pixel were calculated as Δf/f = (f − f0)/f0. The normalized images are shown on a color-coded scale to visualize the relative fluorescence changes (Δf/f) in the cortex.

## 2P imaging data analysis

To correct motion-related artifacts in imaging data, a frame-by-frame alignment algorithm was used to minimize the sum of squared intensity differences between each frame image and a template, which was the average of the selected image frames. To extract fluorescence signals, neurons were visually identified, and drawing regions of interest (ROIs) based on fluorescence intensity was performed. Fluorescence changes (f) were calculated by averaging the corresponding pixel values for each ROI. Relative fluorescence changes Δf/f = (f−f0)/f0 were calculated as $Ca^{2+}$ signals, where the baseline fluorescence f0 was estimated as the 25th percentile of the entire fluorescence recording. To calculate the amplitude of sound-evoked $Ca^{2+}$ transients, we performed automatic $Ca^{2+}$ transient detection based

on threshold criteria regarding peak amplitude and rising rate. The noise level was set to be three times the standard deviation of the baseline (window length: 1 s). The peak amplitude and the rate of rising of the $Ca^{2+}$ signals were calculated to determine whether it was a true transient. The trace of the detected $Ca^{2+}$ transient was first extracted by exponential infinite impulse response (IIR) filtering (window length: 200 ms) and then subtracted from the original signal. The residual fluorescence trace was used as the baseline for the next transient detection, similar to previously published peeling approaches.

## Sound-evoked $Ca^{2+}$ responsiveness

### BBN-evoked $Ca^{2+}$ responsiveness

For data from chronic $Ca^{2+}$ imaging, we tracked the same FOV based on the last training day. We removed the outer ~10% of the image from each ROI to account for edge effects or imaging deviation. The success rate was defined as the fraction of sound-evoked responses during 10 consecutive BBN stimuli. Note that 'Active' neurons showed clear $Ca^{2+}$ transients during the entire recording duration, including spontaneous and BBN-evoked responses. The neurons with a success rate ≥50% were defined as 'Reliable' neurons among active neurons.

### Pure tone-evoked $Ca^{2+}$ responsiveness

The FRA of each ROI was constructed from the 55 average responses to all of the unique frequency-attenuation combinations. The frequency tuning curves were constructed by plotting the average values (and the s.e.m.) of the calcium signal amplitudes from single trials for each frequency tested. If more than one area of contiguous frequency-level combinations remained, the largest one was defined as the FRA. The amplitude of a $Ca^{2+}$ signal was determined as the average value for a period of 200 ms around the peak of the calcium transient. The baseline value was calculated for a period of 100 ms before the onset of the auditory stimulus. Pure tone frequencies that induced response amplitudes higher than half of the maximal response were defined as effective frequencies.

'Irregular' neurons were characterized by exhibiting spontaneous activity patterns that were highly variable and inconsistent in their responses to sound stimulation. These neurons showed no clear or predictable firing pattern. 'Tuned' neurons represented a subset of responsive neurons that demonstrated significant and consistent selectivity for specific auditory stimuli. These neurons exhibited well-defined frequency tuning or preference, responding robustly to certain sound features while showing diminished activity to others. 'Silent' neurons were operationally defined as those that remained completely inactive throughout the entire recording period, which extended beyond 30 min. These neurons showed no detectable spontaneous firing or evoked responses during the experiments. For tuned neurons, the BF was defined as the sound frequency associated with the highest response averaged across all sound levels.

FRAs that exhibited a pattern of decreasing frequency selectivity as sound intensity increased were categorized as V-shaped, reflecting a broadening of the receptive field at higher stimulus levels. This shape indicates that the neuron responds to a wider range of frequencies when the stimulus becomes more intense, suggesting a loss of tuning precision at higher intensities. In contrast, FRAs that maintained consistent frequency selectivity across increasing sound intensities were classified as I-shaped, signifying a stable tuning profile regardless of stimulus amplitude. This pattern implies that the neuron's frequency preference remains sharply defined and resistant to intensity-dependent modulation. Additionally, FRAs that demonstrated responsiveness confined to a narrow range of both sound frequencies and intensities were designated as O-shaped, provided that their peak neural response did not occur at the maximum intensity tested. This classification suggests a limited dynamic range and a preference for intermediate stimulus conditions. Maximum bandwidth ($BW_{max}$) was defined as the maximum FRA width at any level.

To compare the BF heterogeneity, we analyzed the BF distribution in fixed-size analysis windows. The BF heterogeneity was quantified by the IQR of the distribution. For filed IQR, if less than 5 neurons were within a 100 μm radius (including the center neuron), no IQR was calculated (*Schmitt et al., 2023*). For large-scale analysis, imaging regions in separate mice were aligned with each other according to the caudal-to-rostral gradient that was identifiable in the wide-field images.

The MI characterizes the strength of a neuron's intensity tuning. We defined MI as the neuron's response at maximum intensity (80 dB) divided by its maximum response (*Moore and Wehr, 2013*;

*Sutter and Schreiner, 1995*). An MI of 1 indicates no intensity tuning; an MI near 0 indicates very strong intensity tuning.

## Statistics

To compare data between groups, we used the nonparametric Wilcoxon rank-sum test (unpaired) and Wilcoxon signed-rank test (paired) to determine statistical significance (p<0.05) between them. In the text, summarized data are presented as the median\25th–75th percentiles. In the figures, the data presented in the box-and-whisker plot indicate the median (center line), 25th and 75th percentiles (Q1 and Q3), i.e., IQR (box), Q1−1.5×IQR and Q3+1.5 × IQR (whiskers), and all other data with error bars are presented as the mean ± s.e.m.

## Acknowledgements

The authors would like to thank Jia Lou for the cartoon art and figure layout. This work was supported by grants from the National Key R&D Program of China (2021YFA0805000), the National Natural Science Foundation of China (32300937, T2241002, 31925018, 32127801), the Jiangsu Provincial Big Science Facility Initiative (BM2022010), and Guangxi Science and Technology Base & Talents Fund (GUIKE AD22035948). XC is a member of the CAS Center for Excellence in Brain Science and Intelligence Technology.

## Additional information

### Funding

| Funder | Grant reference number | Author |
|---|---|---|
| National Key Research and Development Program of China | 2021YFA0805000 | Xiaowei Chen |
| National Natural Science Foundation of China | 32300937 | Jianxiong Zhang |
| National Natural Science Foundation of China | T2241002 | Jianxiong Zhang |
| National Natural Science Foundation of China | 31925018 | Xiaowei Chen |
| National Natural Science Foundation of China | 32127801 | Xiaowei Chen |
| Jiangsu Provincial Big Science Facility Initiative | BM2022010 | Hongbo Jia |
| Guangxi Science and Technology Base & Talents Fund | GUIKE AD22035948 | Xiaowei Chen |

The funders had no role in study design, data collection and interpretation, or the decision to submit the work for publication.

### Author contributions

Miaoqing Gu, Data curation, Formal analysis, Investigation, Methodology, Writing – original draft; Shanshan Liang, Software, Formal analysis; Jiahui Zhu, Resources, Data curation, Methodology; Ruijie Li, Software, Formal analysis, Methodology; Ke Liu, Investigation, Methodology; Xuanyue Wang, Data curation, Formal analysis, Methodology, Writing – original draft; Frank W Ohl, Visualization, Project administration, Writing – review and editing; Yun Zhang, Formal analysis, Supervision, Validation; Xiang Liao, Formal analysis, Supervision; Chunqing Zhang, Hongbo Jia, Supervision, Validation, Visualization, Writing – review and editing; Yi Zhou, Supervision, Validation, Visualization, Methodology, Writing – review and editing; Jianxiong Zhang, Conceptualization, Formal analysis, Supervision, Funding acquisition, Validation, Visualization, Writing – review and editing; Xiaowei Chen, Conceptualization,

Resources, Supervision, Funding acquisition, Validation, Visualization, Project administration, Writing – review and editing

## Author ORCIDs
Hongbo Jia ⓘ https://orcid.org/0000-0003-1585-2161
Xiaowei Chen ⓘ https://orcid.org/0000-0003-0906-6666

## Ethics
All experiments were approved by the Animal Care Committee of the Third Military Medical University (Approval number: AMUWEC20230061).

Reviewer #1 (Public review): https://doi.org/10.7554/eLife.99989.3.sa1
Reviewer #2 (Public review): https://doi.org/10.7554/eLife.99989.3.sa2
Reviewer #3 (Public review): https://doi.org/10.7554/eLife.99989.3.sa3
Author response https://doi.org/10.7554/eLife.99989.3.sa4

---

# Additional files

## Supplementary files
MDAR checklist

Source data 1. All numerical data used for generating the figures in the manuscript.

## Data availability
The source data are publicly available in the Dryad repository at: https://doi.org/10.5061/dryad.cjsxk-snkm. The source code for this study is openly accessible in a public repository on GitHub at: https://github.com/xieyangshuying/Tonotopy (copy archived at *xieyangshuying, 2025*).

The following dataset was generated:

| Author(s) | Year | Dataset title | Dataset URL | Database and Identifier |
|---|---|---|---|---|
| Gu M, Liang S, Zhu J, Li R, Liu K, Wang K, Ohl F, Zhang Y, Liao X, Zhang C, Jia H, Zhou Y, Zhang J, Chen X | 2025 | Tonotopy is not preserved in a descending stage of auditory cortex | https://doi.org/10.5061/dryad.cjsxksnkm | Dryad Digital Repository, 10.5061/dryad.cjsxksnkm |

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
