## [Editor Report · eLife Assessment]

This revised manuscript presents an **important** characterization of mouse auditory cortex receptive field organization, utilizing two-photon imaging of specific subpopulations. They demonstrate a degradation of tonotopic organization from the input to the output neurons. The strength of the evidence is **convincing**.

---

## [Referee Report · Reviewer #1 (Public review)]

Summary:

In this study, Gu et al., employed novel viral strategies, combined with in vivo two-photon imaging, to map the tone response properties of two groups of cortical neurons in A1 - The thalamocortical recipient (TR neurons) and the corticothalamic (CT neurons). They observed a clear tonotopic gradient among TR neurons but not in CT neurons. Moreover, CT neurons exhibited high heterogeneity of their frequency tuning and broader bandwidth, suggesting increased synaptic integration in these neurons. By parsing out different projecting-specific neurons within A1, this study provides insight into how neurons with different connectivity can exhibit different frequency response-related topographic organization.

Strengths:

This study reveals the importance of studying neurons with projection specificity rather than layer specificity since neurons within the same layer have very diverse molecular, morphological, physiological, and connectional features. By utilizing a newly developed rabies virus CSN-N2c GCaMP-expressing vector, the authors can label and image specifically the neurons (CT neurons) in A1 that project to the MGB. To compare, they used an anterograde trans-synaptic tracing strategy to label and image neurons in A1 that receive input from MGB (TR neurons).

Weaknesses:

- Perhaps as cited in the introduction, it is well known that tonotopic gradient is well preserved across all layers within A1, but I feel if the authors want to highlight the specificity of their virus tracing strategy and the populations that they imaged in L2/3 (TR neurons) and L6 (CT neurons), they should perform control groups where they image general excitatory neurons in the two depths and compare to TR and CT neurons, respectively. This will show that it's not their imaging/analysis or behavioral paradigms that are different from other labs.

- Fig 1D and G, the y-axis is Distance from pia (%). I'm not exactly sure what this means. How does % translate to real cortical thickness?

- For Fig. 2G and H, is each circle a neuron or an animal? Why are they staggered on top of each other on the x-axis? If x-axis is thedistance from caudal to rostral, each neuron should have a different distance? Also,it seems like it's because Fig. 2H has more circles, that's why it has morevariation thus not significant (for example, at 600 or 900um, 2G seems to haveless circles than 2H).

- Similar in Fig 2J and L, why are the circles staggered onthe y-axis now? And is each circle now a neuron or a trial? It seems they havemuch more circles than Fig 2G and 2H. Also I don't think doing a correlation isthe proper stats for this type of plot (this point applies to Fig. 3H and 3J)

What does inter-quartile range of BF (IQRBF, in octaves) imply? What's the interpretation of this analysis? I am confused why TR neurons showhigh IQR in HF areas compared to LF areas mean homogeneity among TR neurons (line 213 - 216). On the same note, how is this different from the BF variability? Isn't higher IQR = tohigher variability?

- Fig. 4A-B, there's no clear critieria on how the authors categorize V, I, and O Shape. The descriptions in the Methods (line 721 - 725) are also very vague.

Comments on revisions:

The authors have addressed all my questions in the previous round.

---

## [Referee Report · Reviewer #2 (Public review)]

Summary:

Gu and Liang et. al investigated how auditory information is mapped and transformed as it enters and exits a auditory cortex. They use anterograde transsynaptic tracers to label and perform calcium imaging of thalamorecipient neurons in A1 and retrograde tracers to label and perform calcium imaging of corticothalamic output neurons. They demonstrate a degradation of tonotopic organization from the input to output neurons.

Strengths:

The experiments appear well executed, well described, and analyzed.

Weaknesses:

(1) Given that the CT and TR neurons were imaged at different depths, the question as to whether not these differences could otherwise be explained by layer-specific differences is still not 100% resolved. Control measurements would be needed either by recording (1) CT neurons upper layers (2) TR in deeper layers (3) non-CT in deeper layers and/or (4) non-TR in upper layers.

(2) What percent of the neurons at the depths being are CT neurons? Similar questions for TR neurons?

(3) V-shaped, I-shaped, or O-shaped is not an intuitively understood nomenclature, consider changing. Further, the x/y axis for Figure 4a is not labeled, so it's not clear what the heat maps are supposed to represent.

(4). Many references about projection neurons and cortical circuits are based on studies from visual or somatosensory cortex. Auditory cortex organization is not necessarily the same as other sensory areas. Auditory cortex references should be used specifically, and not sources reporting on S1, V1.

Comments on revisions:

The authors have fully addressed my concerns.

---

## [Referee Report · Reviewer #3 (Public review)]

Summary:

The authors performed wide-field and 2-photon imaging in vivo in awake head-fixed mice, to compare receptive fields and tonotopic organization in thalamocortical recipient (TR) neurons vs corticothalamic (CT) neurons of mouse auditory cortex. TR neurons were found in all cortical layers while CT neurons were restricted to layer 6. The TR neurons at nominal depths of 200-400 microns have a remarkable degree of tonotopy (as good if not better than tonotopic maps reported by multiunit recordings). In contrast, CT neurons were very heterogenous in terms of their best frequency (BF), even when focusing on the low vs high frequency regions of primary auditory cortex. CT neurons also had wider tuning.

Strengths:

This is a thorough examination using modern methods, helping to resolve a question in the field with projection-specific mapping.

Weaknesses:

There are some limitations due to the methods, and it's unclear what the importance of these responses are outside of behavioral context or measured at single timepoints given the plasticity, context-dependence, and receptive field 'drift' that can occur in cortex.

(1) Probably the biggest conceptual difficulty I have with the paper is comparing these results to past studies mapping auditory cortex topography, mainly due to differences in methods. Conventionally, tonotopic organization is observed for characteristic frequency maps (not best frequency maps), as tuning precision degrades and best frequency can shift as sound intensity increases. The authors used six attenuation levels (30-80 dB SPL) and report that the background noise of the 2-photon scope is <30 dB SPL, which seems very quiet. The authors should at least describe the sound-proofing they used to get the noise level that low, and some sense of noise across the 2-40 kHz frequency range would be nice as a supplementary figure. It also remains unclear just what the 2-photon dF/F response represents in terms of spikes. Classic mapping using single-unit or multi-unit electrodes might be sensitive to single spikes (as might be emitted at characteristic frequency), but this might not be as obvious for Ca2+ imaging. This isn't a concern for the internal comparison here between TR and CT cells as conditions are similar, but is a concern for relating the tonotopy or lack thereof reported here to other studies.

(2) It seems a bit peculiar that while 2721 CT neurons (N=10 mice) were imaged, less than half as many TR cells were imaged (n=1041 cells from N=5 mice). I would have expected there to be many more TR neurons even mouse for mouse (normalizing by number of neurons per mouse), but perhaps the authors were just interested in a comparison data set and not being as thorough or complete with the TR imaging?

(3) The authors definitions of neuronal response type in the methods needs more quantitative detail. The authors state: ""Irregular" neurons exhibited spontaneous activity with highly variable responses to sound stimulation. "Tuned" neurons were responsive neurons that demonstrated significant selectivity for certain stimuli. "Silent" neurons were defined as those that remained completely inactive during our recording period (> 30 min). For tuned neurons, the best frequency (BF) was defined as the sound frequency associated with the highest response averaged across all sound levels." The authors need to define what their thresholds are for 'highly variable', 'significant', and 'completely inactive'. Is best frequency the most significant response, the global max (even if another stimulus evokes a very close amplitude response), etc.

Comments on revisions:

I think the authors misunderstood my point about sound level and characteristic frequency vs best frequency tonotopic maps. Yes, many studies of cortical responses present stimuli at higher intensities than the characteristic frequencies, but as tuning curves widen with sound level, the macroscopic tonotopic organization of primary auditory cortex breaks down at higher intensities. This is why most of the classic studies of tonotopy e.g., from the Merzenich lab generated maps of characteristic frequency. As I mentioned before, this isn't so much of an issue for the authors' comparisons of TR vs CT organization in their own study, but in general, this makes it difficult to compare aspects of spatially-organized tonotopy from imaging studies with the older electrophysiological 'truer' tonotopic maps. That said, this means that CT cells also might be tonotopically organized if the authors had been able to look at lower intensity tuning properties.

---

## [Author Response]

The following is the authors’ response to the original reviews.

**Reviewer #1 (Public Review):**
Summary:In this study, Gu et al. employed novel viral strategies, combined with in vivo two-photon imaging, to map the tone response properties of two groups of cortical neurons in A1. The thalamocortical recipient (TR neurons) and the corticothalamic (CT neurons). They observed a clear tonotopic gradient among TR neurons but not in CT neurons. Moreover, CT neurons exhibited high heterogeneity of their frequency tuning and broader bandwidth, suggesting increased synaptic integration in these neurons. By parsing out different projecting-specific neurons within A1, this study provides insight into how neurons with different connectivity can exhibit different frequency response-related topographic organization.Strengths:This study reveals the importance of studying neurons with projection specificity rather than layer specificity since neurons within the same layer have very diverse molecular, morphological, physiological, and connectional features. By utilizing a newly developed rabies virus CSN-N2c GCaMP-expressing vector, the authors can label and image specifically the neurons (CT neurons) in A1 that project to the MGB. To compare, they used an anterograde trans-synaptic tracing strategy to label and image neurons in A1 that receive input from MGB (TR neurons).Weaknesses:Perhaps as cited in the introduction, it is well known that tonotopic gradient is well preserved across all layers within A1, but I feel if the authors want to highlight the specificity of their virus tracing strategy and the populations that they imaged in L2/3 (TR neurons) and L6 (CT neurons), they should perform control groups where they image general excitatory neurons in the two depths and compare to TR and CT neurons, respectively. This will show that it's not their imaging/analysis or behavioral paradigms that are different from other labs.

We thank the reviewer for these constructive suggestions. As recommended, we have performed control experiments that imaged the general excitatory neurons in superficial layers (shown below), and the results showed a clear tonotopic gradient, which was consistent with previous findings (Bandyopadhyay et al., 2010; Romero et al., 2020; Rothschild et al., 2010; Tischbirek et al., 2019), thereby validating the reliability of our imaging/analysis approach. The results are presented in a new supplemental figure (Figure 2- figure supplementary 3).

Related publications:

(1) Gu M, Li X, Liang S, Zhu J, Sun P, He Y, Yu H, Li R, Zhou Z, Lyu J, Li SC, Budinger E, Zhou Y, Jia H, Zhang J, Chen X. 2023. Rabies virus-based labeling of layer 6 corticothalamic neurons for two-photon imaging in vivo. iScience 26: 106625. DIO: https://doi.org/10.1016/j.isci.2023.106625, PMID: 37250327

(2) Bandyopadhyay S, Shamma SA, Kanold PO. 2010. Dichotomy of functional organization in the mouse auditory cortex. Nat Neurosci 13: 361-8. DIO: https://doi.org/10.1038/nn.2490, PMID: 20118924

(3) Romero S, Hight AE, Clayton KK, Resnik J, Williamson RS, Hancock KE, Polley DB. 2020. Cellular and Widefield Imaging of Sound Frequency Organization in Primary and Higher Order Fields of the Mouse Auditory Cortex. Cerebral Cortex 30: 1603-1622. DIO: https://doi.org/10.1093/cercor/bhz190, PMID: 31667491

(4) Rothschild G, Nelken I, Mizrahi A. 2010. Functional organization and population dynamics in the mouse primary auditory cortex. Nat Neurosci 13: 353-60. DIO: https://doi.org/10.1038/nn.2484, PMID: 20118927

(5) Tischbirek CH, Noda T, Tohmi M, Birkner A, Nelken I, Konnerth A. 2019. In Vivo Functional Mapping of a Cortical Column at Single-Neuron Resolution. Cell Rep 27: 1319-1326 e5. DIO: https://doi.org/10.1016/j.celrep.2019.04.007, PMID: 31042460

Figures 1D and G, the y-axis is Distance from pia (%). I'm not exactly sure what this means. How does % translate to real cortical thickness?

We thank the reviewer for this question. The distance of labeled cells from pia was normalized to the entire distance from pia to L6/WM border for each mouse, according to the previous study (Chang and Kawai, 2018). For all mice tested, the entire distance from pia to L6/WM border was 826.5 ± 23.4 mm (in the range of 752.9 to 886.1).

Related publications:

Chang M, Kawai HD. 2018. A characterization of laminar architecture in mouse primary auditory cortex. Brain Structure and Function 223: 4187-4209. DIO: https://doi.org/10.1007/s00429-018-1744-8, PMID: 30187193

For Figure 2G and H, is each circle a neuron or an animal? Why are they staggered on top of each other on the x-axis? If the x-axis is the distance from caudal to rostral, each neuron should have a different distance? Also, it seems like it's because Figure 2H has more circles, which is why it has more variation, thus not significant (for example, at 600 or 900um, 2G seems to have fewer circles than 2H).

We sincerely appreciate the reviewer’s careful attention to the details of our figures. Each circle in the Figure 2G and H represents an individual imaging focal plane from different animals, and the median BF of some focal planes may be similar, leading to partial overlap. In the regions where overlap occurs, the brightness of the circle will be additive.

Since fewer CT neurons, compared to TR neurons, responded to pure tones within each focal plane, as shown in Figure 2- figure supplementary 2, a larger number of focal planes were imaged to ensure a consistent and robust analysis of the pure tone response characteristics. The higher variance and lack of correlation in CT neurons is a key biological finding, not an artifact of sample size. The data clearly show a wide spread of median BFs at any given location for CT neurons, a feature absent in the TR population.

Similarly, in Figures 2J and L, why are the circles staggered on the y-axis now? And is each circle now a neuron or a trial? It seems they have many more circles than Figure 2G and 2H. Also, I don't think doing a correlation is the proper stats for this type of plot (this point applies to Figures 3H and 3J).

We regret any confusion have caused. In fact, Figure 2 illustrates the tonotopic gradient of CT and TR neurons at different scales. Specifically, Figures 2E-H present the imaging from the focal plane perspective (23 focal planes in Figures 2G, 40 focal planes in Figures 2H), whereas Figures 2I-L provide a more detailed view at the single-cell level (481 neurons in Figures 2J, 491 neurons in Figures 2L). So, Figures 2J and L do indeed have more circles than Figures 2G and H. The analysis at these varying scales consistently reveals the presence of a tonotopic gradient in TR neurons, whereas such a gradient is absent in CT neurons.

We used Pearson correlation as a standard and direct method to quantify the linear relationship between a neuron's anatomical position and its frequency preference, which is widely used in the field to provide a quantitative measure (R-value) and a significance level (p-value) for the strength of a tonotopic gradient. The same statistical logic applies to testing for spatial gradients in local heterogeneity in Figure 3. We are confident that this is an appropriate and informative statistical approach for these data.

What does the inter-quartile range of BF (IQRBF, in octaves) imply? What's the interpretation of this analysis? I am confused as to why TR neurons show high IQR in HF areas compared to LF areas, which means homogeneity among TR neurons (lines 213 - 216). On the same note, how is this different from the BF variability? Isn't higher IQR equal to higher variability?

We thank the reviewer for raising this important point. IQRBF, is a measure of local tuning heterogeneity. It quantifies the diversity of BFs among neighboring neurons. A small IQRBF means neighbors are similarly tuned (an orderly, homogeneous map), while a large IQRBF means neighbors have very different BFs (a disordered, heterogeneous map). (Winkowski and Kanold, 2013; Zeng et al., 2019).

From the BF position reconstruction of all TR neurons (Figures 2I), most TR neurons respond to high-frequency sounds in the high-frequency (HF) region, but some neurons respond to low frequencies such as 2 kHz, which contributes to high IQR in HF areas. This does not contradict our main conclusion, that the TR neurons is significantly more homogeneous than the CT neurons. BF variability represents the stability of a neuron's BF over time, while IQR represents the variability of BF among different neurons within a certain range. (Chambers et al., 2023).

Related publications:

(1) Chambers AR, Aschauer DF, Eppler JB, Kaschube M, Rumpel S. 2023. A stable sensory map emerges from a dynamic equilibrium of neurons with unstable tuning properties. Cerebral Cortex 33: 5597-5612. DIO: https://doi.org/10.1093/cercor/bhac445, PMID: 36418925

(2) Winkowski DE, Kanold PO. 2013. Laminar transformation of frequency organization in auditory cortex. Journal of Neuroscience 33: 1498-508. DIO: https://doi.org/10.1523/JNEUROSCI.3101-12.2013, PMID: 23345224

(3) Zeng HH, Huang JF, Chen M, Wen YQ, Shen ZM, Poo MM. 2019. Local homogeneity of tonotopic organization in the primary auditory cortex of marmosets. Proceedings of the National Academy of Sciences of the United States of America 116: 3239-3244. DIO: https://doi.org/10.1073/pnas.1816653116, PMID: 30718428

Figure 4A-B, there are no clear criteria on how the authors categorize V, I, and O shapes. The descriptions in the Methods (lines 721 - 725) are also very vague.

We apologize for the initial vagueness and have replaced the descriptions in the Methods section. “V-shaped”: Neurons whose FRAs show decreasing frequency selectivity with increasing intensity. “I-shaped”: Neurons whose FRAs show constant frequency selectivity with increasing intensity. “O-shaped”: Neurons responsive to a small range of intensities and frequencies, with the peak response not occurring at the highest intensity level.

To provide better visual intuition, we show multiple representative examples of each FRA type for both TR and CT neurons below. We are confident that these provide the necessary clarity and reproducibility for our analysis of receptive field properties.

**Author response image 1. sa4fig1:** Different FRA types within the dataset of TR and CT neurons. Each row shows 6 representative FRAs from a specific type. Types are V-shaped (‘V'), I-shaped (‘I’), and O-shaped (‘O’). The X-axis represents 11 pure tone frequencies, and the Y-axis represents 6 sound intensities.

**Reviewer #2 (Public Review):**
Summary:Gu and Liang et. al investigated how auditory information is mapped and transformed as it enters and exits an auditory cortex. They use anterograde transsynaptic tracers to label and perform calcium imaging of thalamorecipient neurons in A1 and retrograde tracers to label and perform calcium imaging of corticothalamic output neurons. They demonstrate a degradation of tonotopic organization from the input to output neurons.Strengths:The experiments appear well executed, well described, and analyzed.Weaknesses:(1) Given that the CT and TR neurons were imaged at different depths, the question as to whether or not these differences could otherwise be explained by layer-specific differences is still not 100% resolved. Control measurements would be needed either by recording (1) CT neurons in upper layers, (2) TR in deeper layers, (3) non-CT in deeper layers and/or (4) non-TR in upper layers.

We appreciate these constructive suggestions. To address this, we performed new experiments and analyses.

Comparison of TR neurons across superficial layers: we analyzed our existing TR neuron dataset to see if response properties varied by depth within the superficial layers. We found no significant differences in the fraction of tuned neurons, field IQR, or maximum bandwidth (BWmax) between TR neurons in L2/3 and L4. This suggests a degree of functional homogeneity within the thalamorecipient population across these layers. The results are presented in new supplemental figures (Figure 2- figure supplementary 4).

Necessary control experiments.

(1) CT neurons in upper layers. CT neurons are thalamic projection neurons that only exist in the deeper cortex, so CT neurons do not exist in upper layers (Antunes and Malmierca, 2021).

(2) TR neurons in deeper layers. As we mentioned in the manuscript, due to high-titer AAV1-Cre virus labeling controversy (anterograde and retrograde labelling both exist), it is challenging to identify TR neurons in deeper layers.

(3) non-CT in deeper layers and/or (4) non-TR in upper layers.

To directly test if projection identity confers distinct functional properties within the same cortical layers, we performed the crucial control of comparing TR neurons to their neighboring non-TR neurons. We injected AAV1-Cre in MGB and a Cre-dependent mCherry into A1 to label TR neurons red. We then co-injected AAV-CaMKII-GCaMP6s to label the general excitatory population green. In merged images, this allowed us to functionally image and directly compare TR neurons (yellow) and adjacent non-TR neurons (green). We separately recorded the responses of these neurons to pure tones using two-photon imaging. The results show that TR neurons are significantly more likely to be tuned to pure tones than their neighboring non-TR excitatory neurons. This finding provides direct evidence that a neuron's long-range connectivity, and not just its laminar location, is a key determinant of its response properties. The results are presented in new supplemental figures (Figure 2- figure supplementary 5).

Related publications:

Antunes FM, Malmierca MS. 2021. Corticothalamic Pathways in Auditory Processing: Recent Advances and Insights From Other Sensory Systems. Front Neural Circuits 15: 721186. DIO: https://doi.org/10.3389/fncir.2021.721186, PMID: 34489648

(2) What percent of the neurons at the depths are CT neurons? Similar questions for TR neurons?

We thank the reviewer for the comments. We performed histological analysis on brain slices from our experimental animals to quantify the density of these projection-specific populations. Our analysis reveals that CT neurons constitute approximately 25.47%\22.99%–36.50% of all neurons in Layer 6 of A1. In the superficial layers（L2/3 and L4）, TR neurons comprise approximately 10.66%\10.53%–11.37% of the total neuronal population.

**Author response image 2. sa4fig2:** The fraction of CT and TR neurons. (A) Boxplots showing the fraction of CT neurons. N = 11 slices from 4 mice. (B) Boxplots showing the fraction of TR neurons. N = 11 slices from 4 mice.

(3) V-shaped, I-shaped, or O-shaped is not an intuitively understood nomenclature, consider changing. Further, the x/y axis for Figure 4a is not labeled, so it's not clear what the heat maps are supposed to represent.

The terms "V-shaped," "I-shaped," and "O-shaped" are an established nomenclature in the auditory neuroscience literature for describing frequency response areas (FRAs), and we use them for consistency with prior work. V-shaped: Neurons whose FRAs show decreasing frequency selectivity with increasing intensity. I-shaped: Neurons whose FRAs show constant frequency selectivity with increasing intensity. O-shaped: Neurons responsive to a small range of intensities and frequencies, with the peak response not occurring at the highest intensity level.

(Rothschild et al., 2010). We have included a more detailed description in the Methods.

The X-axis represents 11 pure tone frequencies, and the Y-axis represents 6 sound intensities. So, the heat map represents the FRA of neurons in A1, reflecting the responses for different frequencies and intensities of sound stimuli. In the revised manuscript, we have provided clarifications in the figure legend.

(4) Many references about projection neurons and cortical circuits are based on studies from visual or somatosensory cortex. Auditory cortex organization is not necessarily the same as other sensory areas. Auditory cortex references should be used specifically, and not sources reporting on S1, and V1.

We thank the reviewers for their valuable comments. We have made a concerted effort to ensure that claims about cortical circuit organization are supported by findings specifically from the auditory cortex wherever possible, strengthening the focus and specificity of our discussion.

**Reviewer #3 (Public Review):**
Summary:The authors performed wide-field and 2-photon imaging in vivo in awake head-fixed mice, to compare receptive fields and tonotopic organization in thalamocortical recipient (TR) neurons vs corticothalamic (CT) neurons of mouse auditory cortex. TR neurons were found in all cortical layers while CT neurons were restricted to layer 6. The TR neurons at nominal depths of 200-400 microns have a remarkable degree of tonotopy (as good if not better than tonotopic maps reported by multiunit recordings). In contrast, CT neurons were very heterogenous in terms of their best frequency (BF), even when focusing on the low vs high-frequency regions of the primary auditory cortex. CT neurons also had wider tuning.Strengths:This is a thorough examination using modern methods, helping to resolve a question in the field with projection-specific mapping.Weaknesses:There are some limitations due to the methods, and it's unclear what the importance of these responses are outside of behavioral context or measured at single timepoints given the plasticity, context-dependence, and receptive field 'drift' that can occur in the cortex.(1) Probably the biggest conceptual difficulty I have with the paper is comparing these results to past studies mapping auditory cortex topography, mainly due to differences in methods. Conventionally, the tonotopic organization is observed for characteristic frequency maps (not best frequency maps), as tuning precision degrades and the best frequency can shift as sound intensity increases. The authors used six attenuation levels (30-80 dB SPL) and reported that the background noise of the 2-photon scope is <30 dB SPL, which seems very quiet. The authors should at least describe the sound-proofing they used to get the noise level that low, and some sense of noise across the 2-40 kHz frequency range would be nice as a supplementary figure. It also remains unclear just what the 2-photon dF/F response represents in terms of spikes. Classic mapping using single-unit or multi-unit electrodes might be sensitive to single spikes (as might be emitted at characteristic frequency), but this might not be as obvious for Ca2+ imaging. This isn't a concern for the internal comparison here between TR and CT cells as conditions are similar, but is a concern for relating the tonotopy or lack thereof reported here to other studies.

We sincerely thank the reviewer for the thoughtful evaluation of our manuscript and for your positive assessment of our work.

(1) Concern regarding Best Frequency (BF) vs. Characteristic Frequency (CF)

Our use of BF, defined as the frequency eliciting the highest response averaged across all sound levels, is a standard and practical approach in 2-photon Ca²⁺ imaging studies. (Issa et al., 2014; Rothschild et al., 2010; Schmitt et al., 2023; Tischbirek et al., 2019). This method is well-suited for functionally characterizing large numbers of neurons simultaneously, where determining a precise firing threshold for each individual cell can be challenging.

(2) Concern regarding background noise of the 2-photon setup

We have expanded the Methods section ("Auditory stimulation") to include a detailed description of the sound-attenuation strategies used during the experiments. The use of a custom-built, double-walled sound-proof enclosure lined with wedge-shaped acoustic foam was implemented to significantly reduce external noise interference. These strategies ensured that auditory stimuli were delivered under highly controlled, low-noise conditions, thereby enhancing the reliability and accuracy of the neural response measurements obtained throughout the study.

(3) Concern regarding the relationship between dF/F and spikes

While Ca²⁺ signals are an indirect and filtered representation of spiking activity, they are a powerful tool for assessing the functional properties of genetically-defined cell populations. As you note, the properties and limitations of Ca²⁺ imaging apply equally to both the TR and CT neuron groups we recorded. Therefore, the profound difference we observed—a clear tonotopic gradient in one population and a lack thereof in the other—is a robust biological finding and not a methodological artifact.

Related publications:

(1) Issa JB, Haeffele BD, Agarwal A, Bergles DE, Young ED, Yue DT. 2014. Multiscale optical Ca2+ imaging of tonal organization in mouse auditory cortex. Neuron 83: 944-59. DIO: https://doi.org/10.1016/j.neuron.2014.07.009, PMID: 25088366

(2) Rothschild G, Nelken I, Mizrahi A. 2010. Functional organization and population dynamics in the mouse primary auditory cortex. Nat Neurosci 13: 353-60. DIO: https://doi.org/10.1038/nn.2484, PMID: 20118927

(3) Schmitt TTX, Andrea KMA, Wadle SL, Hirtz JJ. 2023. Distinct topographic organization and network activity patterns of corticocollicular neurons within layer 5 auditory cortex. Front Neural Circuits 17: 1210057. DIO: https://doi.org/10.3389/fncir.2023.1210057, PMID: 37521334

(4) Tischbirek CH, Noda T, Tohmi M, Birkner A, Nelken I, Konnerth A. 2019. In Vivo Functional Mapping of a Cortical Column at Single-Neuron Resolution. Cell Rep 27: 1319-1326 e5. DIO: https://doi.org/10.1016/j.celrep.2019.04.007, PMID: 31042460

(2) It seems a bit peculiar that while 2721 CT neurons (N=10 mice) were imaged, less than half as many TR cells were imaged (n=1041 cells from N=5 mice). I would have expected there to be many more TR neurons even mouse for mouse (normalizing by number of neurons per mouse), but perhaps the authors were just interested in a comparison data set and not being as thorough or complete with the TR imaging?

As shown in the Figure 2- figure supplementary 2, a much higher fraction of TR neurons was "tuned" to pure tones (46% of 1041 neurons) compared to CT neurons (only 18% of 2721 neurons). To obtain a statistically robust and comparable number of tuned neurons for our core analysis (481 tuned TR neurons vs. 491 tuned CT neurons), it was necessary to sample a larger total population of CT neurons, which required imaging from more animals.

(3) The authors' definitions of neuronal response type in the methods need more quantitative detail. The authors state: "Irregular" neurons exhibited spontaneous activity with highly variable responses to sound stimulation. "Tuned" neurons were responsive neurons that demonstrated significant selectivity for certain stimuli. "Silent" neurons were defined as those that remained completely inactive during our recording period (> 30 min). For tuned neurons, the best frequency (BF) was defined as the sound frequency associated with the highest response averaged across all sound levels.". The authors need to define what their thresholds are for 'highly variable', 'significant', and 'completely inactive'. Is best frequency the most significant response, the global max (even if another stimulus evokes a very close amplitude response), etc.

We appreciate the reviewer's suggestions. We have added more detailed description in the Methods.

Tuned neurons: A responsive neuron was further classified as "Tuned" if its responses showed significant frequency selectivity. We determined this using a one-way ANOVA on the neuron's response amplitudes across all tested frequencies (at the sound level that elicited the maximal response). If the ANOVA yielded a p-value < 0.05, the neuron was considered "Tuned”. Irregular neurons: Responsive neurons that did not meet the statistical criterion for being "Tuned" (i.e., ANOVA p-value ≥ 0.05) were classified as "Irregular”. This provides a clear, mutually exclusive category for sound-responsive but broadly-tuned or non-selective cells. Silent neurons: Neurons that were not responsive were classified as "Silent". This quantitatively defines them as cells that showed no significant stimulus-evoked activity during the entire recording session. Best frequency (BF): It is the frequency that elicited the maximal mean response, averaged across all sound levels.

To provide greater clarity, we showed examples in the following figures.

**Author response image 3. sa4fig3:** 

**Reviewer #1 (Recommendations For The Authors):**
(1) A1 and AuC were used exchangeably in the text.

Thank you for pointing out this issue. Our terminological strategy was to remain faithful to the original terms used in the literature we cite, where "AuC" is often used more broadly. In the revised manuscript, we have performed a careful edit to ensure that we use the specific term "A1" (primary auditory cortex) when describing our own results and recording locations, which were functionally and anatomically confirmed.

(2) Grammar mistakes throughout.

We are grateful for the reviewer’s suggested improvement to our wording. The entire manuscript has undergone a thorough professional copyediting process to correct all grammatical errors and improve overall readability.

(3) The discussion should talk more about how/why L6 CT neurons don't possess the tonotopic organization and what are the implications. Currently, it only says 'indicative of an increase in synaptic integration during cortical processing'...

Thanks for this suggestion. We have substantially revised and expanded the Discussion section to explore the potential mechanisms and functional implications of the lack of tonotopy in L6 CT neurons.

Broad pooling of inputs: We propose that the lack of tonotopy is an active computation, not a passive degradation. CT neurons likely pool inputs from a wide range of upstream neurons with diverse frequency preferences. This broad synaptic integration, reflected in their wider tuning bandwidth, would actively erase the fine-grained frequency map in favor of creating a different kind of representation.

A shift from topography to abstract representation: This transformation away from a classic sensory map may be critical for the function of corticothalamic feedback. Instead of relaying "what" frequency was heard, the descending signal from CT neurons may convey more abstract, higher-order information, such as the behavioral relevance of a sound, predictions about upcoming sounds, or motor-related efference copy signals that are not inherently frequency-specific.’

Modulatory role of the descending pathway: The descending A1-to-MGB pathway is often considered to be modulatory, shaping thalamic responses rather than driving them directly. A modulatory signal designed to globally adjust thalamic gain or selectivity may not require, and may even be hindered by, a fine-grained topographical organization.

**Reviewer #2 (Recommendations For The Authors):**
(1) Given that the CT and TR neurons were imaged at different depths, the question as to whether or not these differences could otherwise be explained by layer-specific differences is still not 100% resolved. Control measurements would be needed either by recording (1) CT neurons in upper layers (2) TR in deeper layers (3) non-CT in deeper layers and/or (4) non-TR in upper layers.

We appreciate these constructive suggestions. To address this, we performed new experiments and analyses.

Comparison of TR neurons across superficial layers: we analyzed our existing TR neuron dataset to see if response properties varied by depth within the superficial layers. We found no significant differences in the fraction of tuned neurons, field IQR, or maximum bandwidth (BWmax) between TR neurons in L2/3 and L4. This suggests a degree of functional homogeneity within the thalamorecipient population across these layers.

Necessary control experiments.

(1) CT neurons in upper layers. CT neurons are thalamic projection neurons that only exist in the deeper cortex, so CT neurons do not exist in upper layers (Antunes and Malmierca, 2021).

(2) TR neurons in deeper layers. As we mentioned in the manuscript, due to high-titer AAV1-Cre virus labeling controversy (anterograde and retrograde labelling both exist), it is challenging to identify TR neurons in deeper layers.

(3) non-CT in deeper layers and/or (4) non-TR in upper layers.

To directly test if projection identity confers distinct functional properties within the same cortical layers, we performed the crucial control of comparing TR neurons to their neighboring non-TR neurons. We injected AAV1-Cre in MGB and a Cre-dependent mCherry into A1 to label TR neurons red. We then co-injected AAV-CaMKII-GCaMP6s to label the general excitatory population green. In merged images, this allowed us to functionally image and directly compare TR neurons (yellow) and adjacent non-TR neurons (green). We separately recorded the responses of these neurons to pure tones using two-photon imaging. The results show that TR neurons are significantly more likely to be tuned to pure tones than their neighboring non-TR excitatory neurons. This finding provides direct evidence that a neuron's long-range connectivity, and not just its laminar location, is a key determinant of its response properties.

Related publications:

Antunes FM, Malmierca MS. 2021. Corticothalamic Pathways in Auditory Processing: Recent Advances and Insights From Other Sensory Systems. Front Neural Circuits 15: 721186. DIO: https://doi.org/10.3389/fncir.2021.721186, PMID: 34489648

(3) V-shaped, I-shaped, or O-shaped is not an intuitively understood nomenclature, consider changing. Further, the x/y axis for Figure 4a is not labeled, so it's not clear what the heat maps are supposed to represent.

The terms "V-shaped," "I-shaped," and "O-shaped" are an established nomenclature in the auditory neuroscience literature for describing frequency response areas (FRAs), and we use them for consistency with prior work. V-shaped: Neurons whose FRAs show decreasing frequency selectivity with increasing intensity. I-shaped: Neurons whose FRAs show constant frequency selectivity with increasing intensity. O-shaped: Neurons responsive to a small range of intensities and frequencies, with the peak response not occurring at the highest intensity level.

(Rothschild et al., 2010). We have included a more detailed description in the Methods.

The X-axis represents 11 pure tone frequencies, and the Y-axis represents 6 sound intensities. So, the heat map represents the FRA of neurons in A1, reflecting the responses for different frequencies and intensities of sound stimuli. In the revised manuscript, we have provided clarifications in the figure legend.

(4) Many references about projection neurons and cortical circuits are based on studies from visual or somatosensory cortex. Auditory cortex organization is not necessarily the same as other sensory areas. Auditory cortex references should be used specifically, and not sources reporting on S1, V1.

We thank the reviewers for their valuable comments. We have made a concerted effort to ensure that claims about cortical circuit organization are supported by findings specifically from the auditory cortex wherever possible, strengthening the focus and specificity of our discussion.

**Reviewer #3 (Recommendations For The Authors):**
I suggest showing some more examples of how different neurons and receptive field properties were quantified and statistically analyzed. Especially in Figure 4, but really throughout.

We thank the reviewer for this valuable suggestion. To provide greater clarity, we have added more examples in the following figure.